# Synthesis of Ethyl Pyrimidine-Quinolincarboxylates Selected from Virtual Screening as Enhanced Lactate Dehydrogenase (LDH) Inhibitors

**DOI:** 10.3390/ijms25179744

**Published:** 2024-09-09

**Authors:** Iván Díaz, Sofía Salido, Manuel Nogueras, Justo Cobo

**Affiliations:** Facultad de Ciencias Experimentales, Departamento de Química Inorgánica y Orgánica, Universidad de Jaén, E-23071 Jaén, Spain; idiaz@ujaen.es (I.D.); ssalido@ujaen.es (S.S.); mmontiel@ujaen.es (M.N.)

**Keywords:** hLDHA/hLDHB inhibitors, molecular docking, isatins, ethyl pyrimidine-quinolincarboxylates

## Abstract

The inhibition of the hLDHA (human lactate dehydrogenase A) enzyme has been demonstrated to be of great importance in the treatment of cancer and other diseases, such as primary hyperoxalurias. In that regard, we have designed, using virtual docking screening, a novel family of ethyl pyrimidine-quinolinecarboxylate derivatives (**13**–**18**)(**a**–**d**) as enhanced hLDHA inhibitors. These inhibitors were synthesised through a convergent pathway by coupling the key ethyl 2-aminophenylquinoline-4-carboxylate scaffolds (**7**–**12**), which were prepared by Pfitzinger synthesis followed by a further esterification, to the different 4-aryl-2-chloropyrimidines (**VIII**(**a**–**d**)) under microwave irradiation at 150–170 °C in a green solvent. The values obtained from the hLDHA inhibition were in line with the preliminary of the preliminary docking results, the most potent ones being those with U-shaped disposition. Thirteen of them showed IC_50_ values lower than 5 μM, and for four of them (**16a**, **18b**, **18c** and **18d**), IC_50_ ≈ 1 μM. Additionally, all compounds with IC_50_ < 10 μM were also tested against the hLDHB isoenzyme, resulting in three of them (**15c**, **15d** and **16d**) being selective to the A isoform, with their hLDHB IC_50_ > 100 μM, and the other thirteen behaving as double inhibitors.

## 1. Introduction

The lactate dehydrogenase enzyme (LDH, EC 1.1.1.27) is a tetramer ubiquitously present in the human body, constituted mainly by two different subunits, the M-type (muscle) and the H-type (heart), encoded by the genes *Ldha* and *Ldhb*, respectively. Amongst the five isoforms formed by the combination of those units, the most important ones are the homotetramers hLDHA (M_4_), which are mostly found in the human liver and skeletal muscle, and hLDHB (H_4_), which are mostly found in the human heart and brain [1,2].

One of the main reasons for cancer development is the alteration of cell metabolism. An example of such is the Warburg effect, which promotes cancer growth [3,4,5]. This alteration leads to cells using glycolysis instead of oxidative phosphorylation, resulting in the production of lactate by the hLDHA enzyme. Targeting the hLDHA enzyme could be a promising approach for treating cancer, as it has been the most-researched isoenzyme in this field over the past few decades [6,7,8,9,10,11,12]. In addition to this, the hLDHA enzyme has been associated with various diseases, such as epilepsy [13,14], osteoporosis [15], vascular diseases [16,17], primary hyperoxalurias (PHs) [18,19,20], and arthritis and other inflammatory disorders [21,22,23].

The hLDHB isoenzyme, while not as well documented as hLDHA, is still considered vital in the development of various cancer types [24,25]. It has been associated with aggressive cancer phenotypes [26,27,28] and is known to be overexpressed in breast and lung cancers [27,28,29,30,31,32]. Due to this, the hLDHB enzyme is seen as an ideal target for the prevention and treatment of several types of cancer [26]. In addition to being present in cancer, the hLDHB isoform is also found in different inflammatory [33] and neurodegenerative [34] diseases, and it also acts as a biomarker in ischemic stroke [35].

The inhibition of both hLDHA and hLDHB isoenzymes has been shown to be safe with no severe side effects in humans [24,36]. As a result, new therapeutic approaches are being developed to target these isoenzymes. While other biotechnological approaches, such as using small RNA (siRNA or miRNA), are under development [37,38,39], inhibiting LDH using small molecules has several advantages, including lower costs and an easier administration [40]. However, despite the development of various hLDHA and hLDHB inhibitors (Figure 1) [8,24,26,41,42,43,44], none have been approved for clinical use yet [45].

Having previously developed hLDHA inhibitors [40,44,46,47] and pyrimidine-quinolin/one hybrids [48,49], we decided to optimize their structure in order to improve the potency of our previous pyrimidine-quinolone inhibitors [46]. Pyrimidine and quinoline cores are considered biologically privileged moieties [48,50,51] due to the wide range of activities shown by compounds bearing the pyrimidine [52,53,54,55,56,57,58,59,60] and the quinoline [61,62,63,64,65,66,67,68,69] structures. Some compounds with the pyrimidine and/or quinolone moieties have been reported to be hLDHA inhibitors [70,71,72,73,74]. Thus, our goal is to enhance the potency of these inhibitors based on our previous findings and the biological activity associated with these moieties.

To carry out this objective, we made changes to the quinoline nucleus, including the replacement of the previous oxo group with a carboxylic group (Figure 2). The intention of this change was to enhance the hydrogen bonding ability of the quinolone moiety. The decision to use a carboxylic group was made following reports of hLDHA inhibitors having carboxylic moieties [75,76,77], which are essential for their enzymatic inhibitory activity [78,79,80,81,82].

The new hybrids successfully passed the virtual screening using molecular docking. The Pfitzinger reaction was used to design the chemical synthesis as a strategic step to prepare the ethyl quinoline-4-carboxylate ester intermediates **7**–**12**, which were coupled with 4-aryl-2-chloropyrimidines **VIII**(**a**–**d**). We are pleased to report the synthesis and biological evaluation of twenty novel ethyl pyrimidine-quinoline-4-carboxylate derivatives. Four of them are linked by 1,4- and 1,3-disubstituted aryl moieties in a non-U-shaped disposition (**13**,**14**)(**a**,**b**), and sixteen are linked by a 1,2-disubstituted aryl moiety in a U-shaped disposition (**15**–**18**)(**a**–**d**). Additionally, we have thoroughly evaluated the inhibitory activity of these compounds against both LDH isoforms (hLDHA and hLDHB).

## 2. Results and Discussion

### 2.1. Virtual Screening

The initial virtual screening was conducted through molecular docking using hLDHA-W31 complex (pdb code: **4R68**), as previously documented [46].

A database containing a total of 672 structures of a new family of pyrimidine-quinoline hybrids was created for docking screening. This database was built using a strategic step that involved the Pfitzinger reaction to produce the quinoline-4-carboxylic intermediates. The structures were obtained by combining a variety of substituents to the quinoline, carboxylic derivatives, and aryl residues in the pyrimidine ring. These structures show a linkage between the pyrimidine and quinoline residues through an aminophenyl moiety (see Figure 3).

The analysis of the different substituents (R, X and Ar) was performed as follows: To streamline the process, and based on our previous experience in synthesizing pyrimidine-quinolone hybrids as hLDHA inhibitors [46], all structures were docked in the **W31** site of the enzyme. Once the docking was completed, the output database was analysed according to the raw affinity values and without any further minimization.

Based on the pre-screening results, we found that all structures produced promising results, with raw affinity values being quite good (S ≤ −7 kcal/mol). However, we prioritized the synthesis of structures containing the ester moiety, as they showed the highest number of structures with S ≤ −9 kcal/mol as well as the most similar affinity towards the active site compared to the reference compound **W31** (S = −10.44 kcal/mol).

Furthermore, nearly all of the highest-scoring structures were U-shaped hybrids with the naphth-2-yl, styryl, phenyl, and 4-chlorophenyl moieties in the lipophilic tails. These structures demonstrated strong interactions with Arg^168^ and with the other main amino acid residues responsible for hLDHA activity: His^192^, Asn^137^, and Asp^194^.

A thorough docking study was conducted using the selected structures containing ethyl carboxylate esters. This study involved an analysis of the aforementioned lipophilic tails and the definition of R as H, F, Cl, and Br for the substituents on the quinoline. The docking studies were performed at three sites: the active site (**W31** location), the cofactor site (**NADH**), and the extended site (**W31**-**NADH**).

The results indicate that the only feasible docking position is the active site (**W31** location) because the interactions in the **NADH** site are not strong enough to compete with it. When docked in the extended site, we observed interactions with the same residues as those described in the **W31** site. For more detailed information, please refer to Appendix A.

Focusing on the docking in the **W31** active site, we proceeded to conduct a detailed analysis of all the structures that were initially selected. The best pose for each hybrid was determined using our previously established filtering criteria [46]. The results indicated that only the U-shaped esters (**15**–**18**)(**a**–**d**) advanced to the final stage. We then determined their affinity and energy values involved in the interactions with the main amino acid residues. For detailed information, please refer to Appendix A.

In light of the above, we decided to start synthesizing esters (**13**–**15**)(**a**,**b**) to test the accuracy of the in silico predictions regarding the inhibitory potency of the U-shaped vs. the non-U-shaped compounds. Afterwards, we would determine whether to synthesize the full series of U-shaped hybrids (**15**–**18**)(**a**–**d**), as shown in Figure 4.

### 2.2. Chemistry

After obtaining promising results from virtual screening of quinoline-pyrimidine hybrids containing the ethyl ester group on the quinoline core, we proceeded with their synthesis. In order to accomplish this, we optimized the synthetic strategy to obtain the quinoline residue using the Pfitzinger reaction, which is based on the ring-opening/cyclocondensation sequence of the isatin derivative in its reaction with the acetophenones [83]. Subsequently, the aminophenylquinoline residue would be coupled with the chloropyrimidine fragment through an aromatic nucleophilic substitution (aminolysis from here).

First, we prepared the aminophenylquinolin-4-carboxylic acids intermediates (**1**–**3**) from isatin (**I**) and the corresponding aminoacetophenones (**V**–**VII**) (Figure 1), as reported [84].

It is well known that maintaining a pH of around 6 is crucial in the process of isolating intermediates **1**–**3**, which were obtained in moderate yields (59–80%). If the pH is higher than 6, the carboxylate anion is formed, and at pH lower than 6, the ammonium group is formed, making them soluble in water. This reaction was also extended to other haloisatins **II**–**IV** to produce novel intermediates **4** (F−), **5** (Cl−), and **6** (Br−) with yields of 31%, 50%, and 70%, respectively; in one attempt for **5,** the yield reached 94%. The low yield in the synthesis of compound **4** is attributed to the electronegativity of the fluorine atom and the low reactivity of the amino group in this case, which is in agreement with related reactions between 5-haloisatin (**I**–**IV**) and 4′-aminoacetophenone **V** [85,86].

Once the aminophenylquinolincarboxylic acid intermediates **1**–**6** were prepared, we tested the aminolysis reaction with the 2-(4-aminophenyl)quinoline **1** and 2-chloro-4-(4-chloro)phenylpyrimidine **VIIIa** to adjust the reaction conditions**.**

We initially attempted to carry out the aminolysis reaction using the established conditions with ethanol as the solvent under microwave (MW) irradiation at 120 °C [46,49]. However, the reaction did not proceed as expected to obtain the desired hybrid **A** (see Figure 2). As a result, we investigated the use of different bases (K_2_CO_3_ and Et_3_N), acids (AcOH and HCl), and solvents (H_2_O, EtOH, isopropanol, CH_3_CN, THF, and 1,4-dioxane) together with different heating conditions (reflux or MW) at different temperatures reaching to 160 °C. Despite our efforts, the reaction did not yield the desired product in most cases. In the few instances where it did proceed (EtOH, MW, 150 °C), a complex mixture was obtained, making purification of the desired product **A** very challenging.

After finding that the direct method to synthesize type-**A** carboxylic acid compounds from quinoline-4-carboxylic acid **1** was unsuccessful, we decided to convert the intermediate carboxylic acid **1** into the corresponding ester (**7**) to avoid by-products. This was also to determine if the purification of desired compounds could be easier. Initially, we attempted the classic Fischer’s esterification conditions by carrying out the reaction in ethanol with sulfuric acid; however, this method did not yield good results in terms of purity and isolation simplicity. Furthermore, we also tested this reaction under microwave irradiation with intermediate **1**, but a complex mixture along with the formation of its 4′-*N*-ethyl derivative was observed.

We tested different reaction conditions and found that using thionyl chloride in ethanol, as reported in related studies [87], allowed us to produce compound **7** with a 50% yield. We then applied these conditions to other acids **2**–**6**, resulting in the desired ester derivatives **8**–**12** (see Figure 3). We isolated these compounds through a simple chromatographic separation, obtaining moderate yields of 51–62%, except for compound **12**, which had a yield of 30% due to a more complex mixture.

To set up the conditions for the aminolysis of aminoesters **7**–**12** to be coupled to 2-chloropyrimidines **VIII**(**a**–**d**), we used the reaction between 2-chloro-4-(4-chlorophenyl)pyrimidine **VIIIa** and **7** as a model. The reaction was successfully performed under MW irradiation at 150 °C, yielding **13a** at 85%. This method was then applied to **8**–**9**, resulting in the ester hybrids **14a** and **15a** with good yields (78–81%) in less than 60 min (see Figure 4).

Docking studies indicated that U-shaped esters such as **15a** would have the highest affinities, which the bioassays later confirmed. Once we defined the synthetic pathway for producing the new ethyl pyrimidine-quinolincarboxylic esters, we proceeded with synthesizing the complete series of U-shaped esters. An overview of the defined synthetic pathway is presented in Figure 5. The esterified intermediates **7**–**12** were combined with the corresponding 4-aryl-2-chloropyrimidines **VII**(**a**–**d**) to produce esterified hybrids **13**, **14**(**a**,**b**), and (**15**–**18**)(**a**–**d**) with good yields ranging from 75% to 91%, except for **15d**, which yielded 65%. All of the above structures and their reaction details are provided in Table 1.

In the case of compound **16d**, we obtained a crystalline solid from a DMF solution, enabling us to determine its structure unambiguously through single-crystal X-ray diffraction analysis (Figure 5), which is consistent with the spectroscopic characterization.

### 2.3. hLDH Inhibitory Assays and Preliminary Structure-Activity Relationship

In our study, we initially chose to synthesize and evaluate different ethyl pyrimidine-quinolincarboxylate esters using a spectrofluorometric assay [46] to confirm the reliability of our docking simulations. The results confirmed our predictions, showing that the U-shaped esters **15** had lower IC_50_ values compared to the non-U-shaped esters (**13**–**14**) (see Table 2). Furthermore, among the U-shaped esters, those with the naphth-2-yl group exhibited the best IC_50_ values, with **15b** showing a value of less than 5 μM.

The superior inhibitory effects of U-shaped **15** compared to **13** and **14** are due to the better positioning of **15a** and **15b** in the active site, similar to the reference molecule **W31**. In contrast, the non-U-shaped (**13**,**14**)(**a**,**b**) are located outside of the active site pocket. This information is illustrated in Figure 6.

Due to this finding, we continued to focus only on the evaluation of the U-shaped hybrids **15c**, **15d**, and (**16**–**18**)(**a**–**d**). The results are shown in Table 3, which also includes the data for hLDHB to help with their comparison.

All the hybrid compounds exhibited interesting inhibition data. Thirteen out of sixteen hybrids had IC_50_ values of less than 5 μM. Overall, the hybrids showed good affinity values (S < −9 kcal/mol) and moderate interactions with Arg^168^ (<−5 kcal/mol). Some hybrids also displayed interactions with other key amino acid residues, such as His^192^ and Asp^194^. For specific docking parameters for each U-shaped hybrid, please refer to Appendix A.

Compound **15a** is the only one with an IC_50_ value greater than 10 μM. This can be attributed to its weak interaction with the crucial Arg^168^ amino acid residue. Unlike compound **18a**, which interacts with additional amino acid residues such as His^192^, **15a** does not demonstrate any other interactions beyond Arg^168^.

The majority of hybrids (**15**–**18**)(**a**–**d**) are positioned in the active site of the hLDHA enzyme similar to the reference compound **W31**. This allows them to interact with the different amino acid residues located in the hLDHA active site. This can be seen with **18c** in Figure 7a. Additionally, Figure 7b includes its 2D interaction diagram. For a detailed comparison of the placement of each compound in the hLDHA active site, please refer to Appendix A. 

However, there are some hybrids (**15b**, **16c**, **17c**, and **18a**) which are located differently in the enzyme compared to **W31** (Figure 8). This difference makes them to have stronger interactions with Arg^168^ (**15b**) or to establish another type of interaction with other amino acid residues, as in case of **16c**, **17c**, and **18a**, which are positioned similarly to each other. These differences may be the reason for the good activities displayed by these structures.

After analysing the bioactivity of hybrids (**15**–**18**)(**a**–**d**) against the hLDHA enzyme, we proceeded to evaluate their inhibitory activity towards the other isoform (hLDHB). Consequently, those hybrids having an IC_50_ < 10 μM for the hLDHA enzyme (all hybrids but **15a**) were tested against the B isoform. See Table 3 for the inhibition results (IC_50_).

The results from Table 3 show that three of the hybrids (**15c**, **15d**, and **16d**) selectively inhibit the A isoform, as their IC_50_ values for hLDHB inhibition are greater than 100 μM. On the other hand, the remaining twelve hybrids showed low IC_50_ values for the inhibition of both isoenzymes. This double inhibition is considered an interesting strategy for treating cancer and other proliferative diseases, as it enhances the cytotoxicity of conventional chemotherapeutic drugs. This contributes to demonstrating that the Warburg effect is not solely based on high LDHA expression, as both isozymes need to be targeted to avoid fermentative glycolysis [25,88,89].

In this context, compounds **17a**, **17c**, **18b**, and **18d** exhibited very similar IC_50_ values in both isoforms. The other hybrids (**15b**, **16**(**a**–**c**), **17b**, **17d**, **18a**, and **18c**) showed a slight preference for the A isoform, with an IC_50(B)_/IC_50(A)_ ratio ranging between 1.5 and 4.0.

To further understand the observed bioactivity against the hLDHB isoenzyme, we conducted a detailed docking analysis of the evaluated hybrids **15**(**b**–**d**) and (**16**–**18**)(**a**–**d**).

The reference complex hLDHB-**OXM** (PDB code: **1I0Z**) was chosen, and hybrids **15**(**b**–**d**) and (**16**–**18**)(**a**–**d**) were docked because the active site of hLDHB where oxamate is located is similar to hLDHA. Both complexes were aligned, sharing 75% of residues, with an RMSD of 1.405 Å. They have the central 1,3-cyclohexandione residue in **W31** located in the same place as oxamate and have common interactions with main amino acid residues Arg^168(9)^, His^192(3)^, and Asn^137(8)^.

However, the pocket of the active site in hLDHA is bigger than in hLDHB, with non- interacting Arg^105^ being present outside the pocket (Figure 9) in the latter, affecting the docking analysis and maybe the inhibitory results.

The docking studies in the **OXM** site did not explain the inhibitory results because the size of the hybrids analysed did not fit into the pocket, resulting in positive affinity values, which indicates their repulsion in the active site. Consequently, all hybrids were evaluated in the extended site to see if they could inhibit the enzyme by displacing the NADH.

We selected poses with an RMSD of less than 1.8 Å. Next, we minimized all the selected poses and chose the one with the best or highest number of interactions with the amino acid residues that interact with **NADH**. However, the results did not explain the inhibitory activity of hybrids (**15**–**18**)(**a**–**d**) because the affinity values were not strong enough to displace the **NADH**, and no strong interactions were observed. (Please refer to Appendix A for more details.)

We proceeded to explore other potential allosteric pockets where hybrids (**15**–**18**)(**a**–**d**) could be positioned, as has been reported previously [24]. We found that there is a selective inhibitor of the hLDHB enzyme with an IC_50_ of 42 nM [26] located in an allosteric site of the enzyme (PDB code: **7DBJ**) in the complex hLDHB-**H1U**. 

The selected pose was determined based on the best affinity and energy values in interaction with Glu^214^, which is the main interaction in the reference **H1U** (−40 kcal/mol).

Once the best pose was determined for each structure, we observed that all compounds were positioned in the allosteric site close to the reference **H1U**, with compounds **16a**, **17a**, and **18a** having a very similar pose to **H1U**.

Furthermore, compounds **15d**, **16b**, **16d**, and **17d** exhibited very similar positioning among themselves. Detailed comparisons of their positioning in the hLDHB allosteric site can be found in Appendix A.

Additionally, we observed that the affinity values were quite similar (−8.1 ≤ S ≤ −6.7 kcal/mol) to that of the reported structure (S_H1U_ = −8.667 kcal/mol). These compounds also formed moderate interactions with Glu^214^ and Lys^310^ along with some weaker interactions with Ser^211^. For more detailed information, please refer to Appendix A.

Although the interactions of hybrids (**15**–**18**)(**a**–**d**) with Glu^214^ were not as strong, as in the case of reference **H1U**, they were of a similar magnitude to those demonstrated with different amino acid residues in the case of hLDHA. Considering the IC_50_ values obtained for both isoforms, these findings provide a reasonable explanation for the experimental results.

Additionally, compounds **15c**, **15d**, and **16d** did not interact with either of the two possible Lys^310^ residues (Figure 10a), which could explain their selectivity towards the A isoform. Therefore, this interaction seems crucial in these hybrid systems for inhibiting the hLDHB enzyme, either with the Lys^310^ residue of one chain or another.

It is worth noting that all of the evaluated hybrids interacted with the Glu^214^ amino acid residue of chain C through the amino group directly linked to the pyrimidine core. In contrast, the interaction with Lys^310^ occurred with either of the residues from chain A or C, typically through the ester group or the nitrogen heteroatom belonging to the pyrimidine or quinoline ring, as illustrated with compounds **15a** and **16a** in Figure 10b.

## 3. Materials and Methods

### 3.1. General

All chemicals and solvents were purchased from Sigma-Aldrich (Merck KGaA, Darmstadt, Germany) unless stated otherwise.

Melting points were collected using a Barnstead Electrothermal 9100 melting point apparatus (Dubuque, IA, USA), and the acquired data are uncorrected. IR spectra were recorded on a Fourier Bruker Tensor 27 spectrophotometer (Leipzig, Germany) using the ATR dura Sample IR accessory. 

NMR spectra were recorded in Bruker Avance NEO 400 spectrometer (Karlsruhe, Germany) at 400 MHz (^1^H) and 100 MHz (^13^C) at 298 K (and 393 K, if specified) using as solvent CDCl_3_ and DMSO-*d*_6_ and, as the internal reference, tetramethylsilane (0 ppm), or the residual ^1^H/^13^C solvent signals, that is, 7.26/77.16 for CDCl_3_ and 2.50/39.52 for DMSO-*d*_6_. DEPT-135 and 2D-NMR (HSQC, HMBC, and COSY) experiments were used for the assignment of carbon and hydrogen signals. Chemical shifts (δ) are given in ppm, and coupling constants (*J*) are given in Hz. The following abbreviations are used for multiplicities: s = singlet, d = doublet, t = triplet, q = quartet, m = multiplet, ps = pseudosinglet, pd = pseudodoublet, and pt = pseudotriplet. Full assignation of all hydrogen and carbon atoms in the structures is shown in SI.

The mass spectra were recorded on a Thermo model DSQ II spectrometer (Waltham, MA, USA) (equipped with a direct inlet probe) and operating at 70 eV. HPLC-HRMS data were obtained on an Agilent Technologies Q-TOF 6530B coupled to an HPLC Agilent-1260 Infinity (Santa Clara, CA, USA), equipped with a Kinetex C18 column (2.1 mm × 50 mm × 2.6 μm) PN 00B-4462-AN using the following HPLC method: flow, 0.4 mL/min; elution gradient, 0–5 min from acetonitrile/water 10% (0.1% formic acid) to acetonitrile 100% (0.1% formic acid) plus 3 additional minutes at that concentration; ionization method, electrospray ionization (ESI+); acquisition software, MassHunter LC/MS Data Acquisition 6200 series TOD/6500 series Q-TOF; version, B.06.01 (Build 6.01.6172 SP1). 

The single-crystal X-ray diffraction measurements for a crystal suitable for **16d** were collected on a Hampton Research Mounted CryoloopTM with a Diffractometer Bruker D8 Venture (APEX 4) (Madison, WI, USA), monochromator multilayer mirror, CCD rotation images, thick slices φ and θ scans, and with a Mo INCOATEC high-brilliance microfocus sealed tube as an X-ray source (λ = 0.71073 Å). Data collection: APEX4 v2021.10-0; cell refinement: SAINT V8.40B [90,91,92]; data reduction: SAINT V8.37A; 16 program(s) used to solve structure: SHELXT-2014/5 [93], program(s) used to refine structure: SHELXL-2019/1 [94], software used to prepare material for publication: Wingx 2018.2 [95] and Mercury 3.10.3 [96].

All the equipment used in the spectroscopic and spectrometric analysis belong to “Centro de Instrumentación Científico y Técnico”, (CICT) in “Universidad de Jaén” (UJA).

All reactions were TLC monitored on a 0.2 mm pre-coated aluminium plate of silica gel (Merck 60 F_254_), and spots were visualized by UV irradiation (254 nm).

Purifications of synthesised compounds were performed by flash column chromatography (FCC) using Silica gel 60 (particle size 0.040–0.063 mm) (Merck KGaA, Darmstadt, Germany).

All reagents were purchased from commercial sources and used without further purification. All starting materials were weighed and handled in air at room temperature. Precursor 4-aryl-2-chloropyrimidines **VIII**(**a**–**d**) were prepared according to the reported procedure [48].

### 3.2. Chemistry

#### 3.2.1. General Procedure for the Synthesis of 2-Aminophenylquinoline-4-carboxylic Acids (**1**–**6**) [97]

Potassium hydroxide (3 mmol) and water (0.10 mL) were added to a round-bottom flask. Then, ethanol (10 mL), the corresponding isatin derivate **I**–**IV** (1 mmol), and aminoacetophenone **V**–**VII** (1.2 mmol) were added sequentially, and the mixture was heated to reflux under stirring. After 24 h, the solvent was removed under vacuum. Then, water (≈10 mL) was added, and the crude was washed with Et_2_O (3 × 15 mL). The aqueous phase was treated with dilute hydrochloric acid (5%) to pH ≈ 5–6, and the precipitated solid was filtered off and washed with cold water and Et_2_O. Compounds (**1**–**6**) were used without further purification.

2-(4-Aminophenyl)quinoline-4-carboxylic acid (**1**).From isatin **I** and 4-aminoacetophenone **V**. Red solid (80%) M.p. (193–195) °C. R_f_ DCM:MeOH (9:1): 0.07. ^1^H NMR (400 MHz, DMSO-d_6_) δ 8.57 (d, *J* = 8.3 Hz, 1H), 8.31 (s, 1H), 8.03 (m, 3H), 7.76 (pt, *J* = 7.2 Hz, 1H), 7.58 (pt, *J* = 7.2 Hz, 1H), 6.71 (d, *J* = 8.6 Hz, 2H). ^13^C NMR (100 MHz, DMSO-d_6_) δ 167.9, 156.2, 151.0, 148.5, 137.0, 129.9, 129.2, 128.4, 126.5, 125.3, 124.9, 122.7, 118.3, 113.8.Structure appears as reported [84] and patented [85], but no melting point is given.2-(3-Aminophenyl) quinoline-4-carboxylic acid (**2**).From isatin **I** and 3-aminoacetophenone **VI**. White solid (59%) M.p. (179–181) °C. R_f_ DCM:MeOH (9:1): 0.02. ^1^H NMR (400 MHz, DMSO-d_6_) δ 8.68 (d, *J* = 8.4 Hz, 1H), 8.36 (s, 1H), 8.13 (d, *J* = 8.3 Hz, 1H), 7.84 (pt, *J* = 7.5 Hz, 1H), 7.68 (pt, *J* = 7.5 Hz, 1H), 7.54 (s, 1H), 7.40 (d, *J* = 7.5 Hz, 1H), 7.21 (t, *J* = 7.7 Hz, 1H), 6.73 (d, *J* = 7.5 Hz, 1H). ^13^C NMR (100 MHz, DMSO d_6_) δ 167.7, 156.5, 149.2, 148.4, 138.5, 137.1, 130.2, 129.7, 129.6, 127.6, 125.4, 123.5, 119.3, 115.7, 114.9, 112.4.Structure appears as patented [84], but no characterization data are reported.2-(2-Aminophenyl)quinoline-4-carboxylic acid (**3**).From isatin **I** and 2-aminoacetophenone **VII**. Yellow solid (74%) M.p. (227–229) °C. R_f_ DCM:MeOH, 9:1: 0.17. ^1^H NMR (400 MHz, DMSO-*d_6_* at 393 K) δ 8.64 (d, *J* = 8.3 Hz, 1H), 8.37 (s, 1H), 8.14 (d, *J* = 8.3 Hz, 1H), 7.89–7.78 (m, 2H), 7.68 (pt, *J* = 7.3 Hz, 1H), 7.21 (t, *J* = 7.1 Hz, 1H), 6.92 (d, *J* = 8.1 Hz, 1H), 6.75 (t, *J* = 7.2 Hz, 1H). ^13^C NMR (100 MHz, DMSO) δ 167.6, 158.2, 147.5, 147.1, 137.1, 130.7, 130.2, 129.6, 129.0, 127.4, 125.3, 122.3, 120.8, 119.3, 117.4, 116.7.Structure appears as patented [84], but no characterization data are reported.2-(2-Aminophenyl)-6-fluoroquinoline-4-carboxylic acid (**4**).From 5-fluoroisatin **II** and 2-aminoacetophenone **VII**. Yellow solid (31%) M.p. (210–212) °C. R_f_ DCM:MeOH (9:1): 0.11. ^1^H NMR (400 MHz, DMSO-d_6_, at 393 K) δ 8.42 (dd, *^3^J_HF_* = 10.2 Hz, *J* = 2.9 Hz, 1H), 8.40 (s, 1H), 8.16 (dd, *J* = 9.2, *^4^J_HF_* = 5.7 Hz, 1H), 7.72 (dd, *J* = 8.0, 1.3 Hz, 1H), 7.67 (ddd, *J* = 9.2, 2.9 Hz, *^3^J_HF_* = 8.3 Hz, 1H), 7.18 (ddd, *J* = 8.3, 7.2, 1.3 Hz, 1H), 6.89 (dd, *J* = 8.3, 1.3 Hz, 1H), 6.72 (ddd, *J* = 8.0, 7.2, 1.3 Hz, 1H). ^13^C NMR (100 MHz, DMSO-d_6_) δ 166.3, 159.8 (d, *^1^J_CF_* = 245.4 Hz), 157.4 (d, *^6^J_C_*_F_ = 2.5 Hz), 147.7, 144.1, 136.2 (d, *^4^J_CF_* = 5.5 Hz), 131.0 (d, *^3^J_CF_* = 10.9 Hz), 129.7, 128.7, 122.8 (d, *^3^J_CF_* = 10.9 Hz), 121.2, 119.0 (d, *^2^J_CF_* = 25.0 Hz), 118.8, 116.4, 115.5, 108.5 (d, *^2^J_CF_* = 25.0 Hz). IR (ATR, cm^−1^): (3500–1900) (broad NH_2_ along with COOH signal), 1712 (C=O), 1549, 1234, 749, 468. EI MS (70 eV): m/z (%): 282 (M+, 78), 281 (62), 237 (100), 236 (64). HRMS (ESI-QTOF) (M+H) calc. for C_16_H_11_FN_2_O_2_: 283.0877 found: 283.0879. HRMS (ESI-QTOF) (M-H) calc. for C_16_H_11_FN_2_O_2_: 281.0732 found: 281.0732.2-(2-Aminophenyl)-6-chloroquinoline-4-carboxylic acid (**5**).From 5-chloroisatin **III** and 2-aminoacetophenone **VII**. Brown solid (94%) M.p. (215–217) °C. R_f_ DCM:MeOH (9:1): 0.15. ^1^H NMR (400 MHz, DMSO-d_6_, at 393 K) δ 8.78 (d, *J* = 2.4 Hz, 1H), 8.37 (s, 1H), 8.09 (d, *J* = 9.0 Hz, 1H), 7.76 (dd, *J* = 9.0, 2.4 Hz, 1H), 7.73 (dd, *J* = 8.2, 1.2 Hz, 1H), 7.18 (ddd, *J* = 8.2, 7.0, 1.2 Hz, 1H), 6.88 (dd, *J* = 8.2, 1.2 Hz, 1H), 6.72 (ddd, *J* = 8.2, 7.0, 1.2 Hz, 1H). ^13^C NMR (100 MHz, DMSO-d_6_) δ 166.4, 158.4, 147.9, 145.3, 137.5, 130.9, 130.1, 129.9, 129.4, 128.7, 124.0, 123.0, 120.9, 118.8, 116.5, 115.5. IR (ATR, cm^−1^): 3460 (NH_2_), 3330 (NH_2_), (3200–1900) (broad COOH signal), 1720 (C=O), 1548, 1320, 760, 666. EI MS (70 eV): m/z (%): 298 (M+, 88), 297 (77), 253 (100), 252 (47). HRMS (ESI-QTOF) (M+H) calc. for C_16_H_11_ClN_2_O_2_: 299.0582 found: 299.0584. HRMS (ESI-QTOF) (M-H) calc. for C_16_H_11_ClN_2_O_2_: 297.0436 found: 297.0437.2-(2-Aminophenyl)-6-bromoquinoline-4-carboxylic acid (**6**).From 5-bromoisatin **IV** and 2-aminoacetophenone **VII**. Brown solid (70%) M.p. (243–245) °C. R_f_ DCM:MeOH (9:1): 0.15. ^1^H NMR (400 MHz, DMSO-d_6_, at 393 K) δ 8.93 (d, *J* = 2.2 Hz, 1H), 8.37 (s, 1H), 8.02 (d, *J* = 8.9 Hz, 1H), 7.88 (dd, *J* = 8.9, 2.2 Hz, 1H), 7.74 (dd, *J* = 8.0, 1.2 Hz, 1H), 7.18 (ddd, *J* = 8.3, 7.0, 1.2 Hz, 1H), 6.88 (pd, *J* = 8.3 Hz, 1H), 6.72 (ddd, *J* = 8.0, 7.0, 1.2 Hz, 1H). ^13^C NMR (100 MHz, DMSO-d_6_) δ 166.3, 158.5, 147.9, 145.5, 136.7, 132.1, 130.2, 129.9, 128.8, 127.2, 123.3, 121.0, 119.5, 118.7, 116.5, 115.5. IR (ATR, cm^−1^): (3500–1900) (broad, NH_2_ along with COOH signal), 1712 (C=O), 1544, 1452, 1371, 1319, 758, 661. EI MS (70 eV): m/z (%): 342 (M+, 95), 340 (82), 299 (76), 298 (52), 297 (100), 218 (87), 217 (50). HRMS (ESI-QTOF) (M+H) calc. for C_16_H_11_BrN_2_O_2_: 343.0077 found: 343.0076. HRMS (ESI-QTOF) (M-H) calc. for C_16_H_11_BrN_2_O_2_: 340.9931 found: 340.9933.

#### 3.2.2. General Procedure for the Synthesis of Ethyl 2-(2-Aminophenyl)quinoline-4-carboxylates (**7**–**12**)

The corresponding 2-aminophenylquinoline-4-carboxylic acids (**1**–**6**) (1 mmol) were dissolved in ethanol (10 mL/mmol) and cooled down to 0 °C. Then, thionyl chloride (2 mmol) was added dropwise under argon atmosphere. Next, the mixture was allowed to reflux, and when the reaction was completed (24 h, TLC monitored using DCM:MeOH, 9:1), the solvent was removed under vacuum, and water (15 mL) was added. The mixture was neutralized with solid NaHCO_3_ to pH = 7 and extracted with ethyl acetate (3 × 20 mL). The organic phases were joined, dried over anhydrous sodium sulphate, and the solvent was removed under vacuum. The desired compounds were purified by FCC using Hex:AcOEt (9:1) as eluent.

Ethyl 2-(4-aminophenyl)quinoline-4-carboxylate (**7**).From compound **1**. Yellow solid (50%) M.p. (143–145) °C [Lit [97] 418 K]. R_f_ Hex:AcOEt (7:3): 0.26. ^1^H NMR (400 MHz, CDCl_3_) δ 8.70–8.65 (m, 1H), 8.31 (s, 1H), 8.18–8.13 (m, 1H), 8.10–8.04 (m, 2H), 7.72 (ddd, *J* = 8.4, 6.9, 1.4 Hz, 1H), 7.56 (ddd, *J* = 8.4, 6.9, 1.4 Hz, 1H), 6.86–6.79 (m, 2H), 4.54 (q, *J* = 7.1 Hz, 2H), 3.93 (s, 2H), 1.50 (t, *J* = 7.1 Hz, 3H). ^13^C NMR (100 MHz, CDCl_3_) δ 166.8, 156.7, 149.4, 148.3, 135.9, 130.1, 129.8, 129.2, 128.9, 127.1, 125.5, 123.6, 119.8, 115.3, 62.0, 14.5.Ethyl 2-(3-aminophenyl)quinoline-4-carboxylate (**8**).From compound **2**. Yellow oil (52%) R_f_ Hex:AcOEt (7:3): 0.37. ^1^H NMR (400 MHz, CDCl_3_) δ 8.73 (ddd, *J* = 8.4, 1.4, 0.6 Hz, 1H), 8.35 (s, 1H), 8.21 (ddd, *J* = 8.5, 1.4, 0.6 Hz, 1H), 7.76 (ddd, *J* = 8.4, 6.9, 1.4 Hz, 1H), 7.62 (ddd, *J* = 8.4, 6.9, 1.4 Hz, 1H), 7.60–7.58 (m, 1H), 7.52 (ddd, *J* = 7.8, 1.7, 1.0 Hz, 1H), 7.36–7.30 (m, 1H), 6.82 (ddd, *J* = 7.8, 2.4, 1.0 Hz, 1H), 4.55 (q, *J* = 7.1 Hz, 2H), 3.85 (s, 2H), 1.50 (t, *J* = 7.1 Hz, 3H). 13C NMR (101 MHz, CDCl_3_) δ 166.6, 157.0, 149.3, 147.2, 140.1, 136.1, 130.4, 129.98, 129.96, 127.8, 125.5, 124.2, 120.6, 118.0, 116.7, 114.1, 62.0, 14.5.Structure appears as patented [98], but no characterization data are reported.Ethyl 2-(2-aminophenyl)quinoline-4-carboxylate (**9**).From compound **3**. Orange solid (58%) M.p. (71–73) °C R_f_ Hex:AcOEt (9:1): 0.22. ^1^H NMR (400 MHz, DMSO-d_6_) δ 8.51 (d, *J* = 8.3 Hz, 1H), 8.36 (s, 1H), 8.13 (d, *J* = 8.3 Hz, 1H), 7.87–7.76 (m, 2H), 7.67 (pt, *J* = 7.4 Hz, 1H), 7.27–7.03 (m, 3H), 6.87 (pd, *J* = 8.3 Hz, 1H), 6.69 (pt, *J* = 7.3 Hz, 1H), 4.48 (q, *J* = 7.1 Hz, 2H), 1.41 (t, *J* = 7.1 Hz, 3H). ^13^C NMR (100 MHz, DMSO-d_6_) δ 165.9, 158.3, 148.8, 147.0, 136.1, 130.8, 130.3, 129.5, 129.0, 127.5, 124.9, 121.9, 120.7, 118.3, 116.9, 115.8, 61.9, 14.1. IR (ATR, cm^−1^): 3429 (NH), 3271 (N-H--N_Q_, H-bond), 1703 (C=O), 1593, 1544, 1472, 1250, 758, 656. EI MS (70 eV): m/z (%): 292 (M+, 28), 219 (31), 91 (100). HRMS (ESI-QTOF) (M+H) calc. for C_18_H_16_N_2_O_2_: 293.1285 found: 293.1286.Ethyl 2-(2-aminophenyl)-6-fluoroquinoline-4-carboxylate (**10**).From compound **4**. Orange solid (62%) M.p. (110–112) °C. R_f_ Hex:AcOEt (7:3): 0.46. ^1^H NMR (400 MHz, DMSO-d_6_) δ 8.41 (s, 1H), 8.29 (dd, *^3^J_HF_* = 10.8 Hz, *J* = 2.9 Hz, 1H), 8.21 (dd, *J* = 9.2 Hz, *^4^J_HF_* = 5.7 Hz, 1H), 7.78–7.71 (m, 2H), 7.18 (ddd, *J* = 8.3. 7.0, 1.2 Hz,, 1H), 7.05 (s, 2H), 6.87 (dd, *J* = 8.3, 1.2 Hz, 1H), 6.69 (ddd, *J* = 8.0, 7.0, 1.2 Hz, 1H), 4.47 (q, *J* = 7.1 Hz, 2H), 1.41 (t, *J* = 7.1 Hz, 3H). ^13^C NMR (100 MHz, DMSO-d_6_) δ 165.4, 160.4 (d, *^1^J_CF_* = 245.4 Hz), 157.8 (d, *^6^J_CF_* = 3.0 Hz), 148.6, 144.4, 134.9 (d, *^4^J_CF_* = 5.5 Hz), 131.9 (d, *^3^J_CF_* = 9.6 Hz), 130.8, 129.4, 122.8 (d, *^3^J_CF_* = 11.0 Hz), 121.9, 120.1 (d, *^2^J_CF_* = 25.0 Hz), 118.1, 116.9, 115.8, 108.9 (d, *^2^J_CF_* = 25.0 Hz), 62.0, 14.0. IR (ATR, cm^−1^): 3429 (NH), 3267 (N-H--N_Q_, H-bond), 1702 (C=O), 1595, 1545, 1475, 1227, 740, 645. EI MS (70 eV): m/z (%): 310 (M+, 84), 309 (67), 237 (100), 236 (46). HRMS (ESI-QTOF) (M+H) calc. for C_18_H_15_FN_2_O_2_: 311.1193 found: 311.1190.Ethyl 2-(2-aminophenyl)-6-chloroquinoline-4-carboxylate (**11**).From compound **5**. Orange solid (51%) M.p. (145–147) °C. R_f_ Hex:AcOEt (9:1): 0.20. ^1^H NMR (400 MHz, DMSO-d_6_) δ 8.60 (d, *J* = 2.2 Hz, 1H), 8.41 (s, 1H), 8.15 (d, *J* = 9.0 Hz, 1H), 7.82 (dd, *J* = 9.0, 2.2 Hz, 1H), 7.77 (d, *J* = 8.0 Hz, 1H), 7.19 (pt, *J* = 7.8 Hz, 1H), 7.13 (broad s, 2H), 6.87 (d, *J* = 8.3 Hz, 1H), 6.68 (pt, *J* = 7.6 Hz, 1H), 4.47 (q, *J* = 7.1 Hz, 2H), 1.41 (t, *J* = 7.1 Hz, 3H). ^13^C NMR (100 MHz, DMSO-d_6_) δ 165.3, 158.7, 148.8, 145.6, 134.6, 132.0, 131.1, 131.0, 130.6, 129.5, 123.9, 122.7, 121.9, 117.8, 117.0, 115.8, 62.0, 14.0. IR (ATR, cm^−1^): 3414 (NH), 3231 (N-H--NQ, H-bond), 1699 (C=O), 1587, 1463, 1270, 729, 677. EI MS (70 eV): m/z (%): 326 (M+, 66), 325 (56), 253 (100), 66 (77). HRMS (ESI-QTOF) (M+H) calc. for C_18_H_15_ClN_2_O_2_: 327.0895 found: 327.0900.Ethyl 2-(2-aminophenyl)-6-bromoquinoline-4-carboxylate (**12**).From compound **6**. Orange solid (30%) M.p. (148–150) °C. R_f_ Hex:AcOEt (9:1): 0.17. ^1^H NMR (400 MHz, DMSO-d_6_) δ 8.77 (ps, 1H), 8.41 (s, 1H), 8.08 (d, *J* = 8.8 Hz, 1H), 7.94 (d, *J* = 8.8 Hz, 1H), 7.78 (d, *J* = 8.0 Hz, 1H), 7.19 (pt, *J* = 7.8 Hz, 1H), 7.14 (s, 2H), 6.87 (d, *J* = 8.3 Hz, 1H), 6.69 (pt, *J* = 7.6 Hz, 1H), 4.47 (q, *J* = 7.1 Hz, 2H), 1.41 (t, *J* = 7.1 Hz, 3H). ^13^C NMR (100 MHz, DMSO-d_6_) δ 165.3, 158.8, 148.9, 145.8, 134.5, 133.2, 131.2, 131.0, 129.5, 127.1, 123.1, 121.9, 120.7, 117.8, 117.0, 115.8, 62.0, 14.0. IR (ATR, cm^−1^): 3417 (NH), 3231 (N-H--NQ, H-bond), 1700 (C=O), 1584, 1462, 729, 668. EI MS (70 eV): m/z (%): 370 (M+, 36), 299 (76), 297 (100), 218 (50), 84 (49), 66 (83). HRMS (ESI-QTOF) (M+H) calc. for C_18_H_15_BrN_2_O_2_: 371.0390 found: 371.0391.

#### 3.2.3. General Procedure for the Synthesis of Ethyl 2-(2-((4-Arylpyrimidin-2-yl)amino) phenyl)quinoline-4-carboxylates (**13**–**18**)(**a**–**d**)

The corresponding ethyl 2-(2-aminophenyl)quinoline-4-carboxylate **7**–**12** (1.05 mmol) and 4-aryl-2-chloropyrimidine **VIII**(**a**–**d**) (1 mmol) were dissolved in ethanol (1.5 mL), and then the mixture was subjected to microwave irradiation at 150 °C with a setting of 250 psi and 250 W. Once the reaction was completed (TLC monitored using Hex:AcOEt, 8:2), the precipitated solid was filtered off and washed with ethanol; if no solid was formed, then the solvent was removed under reduced pressure, and the residue was dissolved with ethyl acetate (10 mL) and water (10 mL) was added. The aqueous phase was washed several times with ethyl acetate (3 × 10 mL). The organic phases were joined, dried over anhydrous sodium sulphate, and the solvent was removed under vacuum. If needed, compounds (**13**–**18**)(**a**–**d**) were further purified by FCC.

Ethyl 2-(4-((4-(4-chlorophenyl)pyrimidin-2-yl)amino)phenyl)quinoline-4-carboxylate (**13a**).From compounds **7** and **VIIIa**. Reaction time: 50 min. Orange solid (85%) M.p. (192–194) °C. R_f_ Hex:AcOEt (7:3): 0.39. ^1^H NMR (400 MHz, DMSO-d_6_) δ 10.13 (s, 1H), 8.63 (d, *J* = 5.2 Hz, 1H), 8.51 (d, *J* = 8.5 Hz, 1H), 8.45 (s, 1H), 8.30 (d, *J* = 8.7 Hz, 2H), 8.23 (d, *J* = 8.5 Hz, 2H), 8.19 (d, *J* = 8.4 Hz, 1H), 8.06 (d, *J* = 8.7 Hz, 2H), 7.85 (pt, *J* = 7.7 Hz, 1H), 7.68 (pt, *J* = 7.7 Hz, 1H), 7.64 (d, *J* = 8.5 Hz, 2H), 7.50 (d, *J* = 5.2 Hz, 1H), 4.50 (q, *J* = 7.1 Hz, 2H), 1.44 (t, *J* = 7.1 Hz, 3H). ^13^C NMR (100 MHz, DMSO-d_6_) δ 165.9, 162.6, 159.8, 159.2, 155.4, 147.7, 142.8, 137.0, 135.9, 135.3, 130.6, 129.8, 129.1, 129.0, 128.8, 128.0, 127.7, 125.1, 122.8, 119.0, 118.8, 108.5, 62.0, 14.1. IR (ATR, cm^−1^): 3214 (NH), 1684 (C=O), 1533, 1500, 1387, 1215, 1069, 742. EI MS (70 eV): m/z (%): 480 (M+, 100), 451 (32), 226 (16). HRMS (ESI-QTOF) (M+H) calc. for C_28_H_21_ClN_4_O_2_: 481.1426 found: 481.1428.Ethyl 2-(4-((4-(naphthalen-2-yl)pyrimidin-2-yl)amino)phenyl)quinoline-4-carboxylate (**13b**).From compounds **7** and **VIIIb**. Reaction time: 60 min. Orange solid (86%) M.p. (194–196) °C. R_f_ Hex:AcOEt (7:3): 0.27. ^1^H NMR (400 MHz, DMSO-d_6_) δ 10.13 (s, 1H), 8.80 (s, 1H), 8.66 (d, *J* = 5.2 Hz, 1H), 8.51 (dd, *J* = 8.5, 1.0 Hz, 1H), 8.46 (s, 1H), 8.37–8.29 (m, 3H), 8.18–8.08 (m, 5H), 8.02–7.98 (m, 1H), 7.84 (ddd, *J* = 8.4, 7.0, 1.2 Hz, 1H), 7.67 (ddd, *J* = 8.5, 7.0, 1.2 Hz, 1H), 7.65–7.59 (m, 3H), 4.50 (q, *J* = 7.1 Hz, 2H), 1.44 (t, *J* = 7.1 Hz, 3H). ^13^C NMR (100 MHz, DMSO-d_6_) δ 166.0, 163.8, 159.9, 159.0, 155.5, 147.9, 142.8, 136.8, 134.2, 133.9, 132.8, 130.5, 130.0, 129.2, 129.0, 128.6, 128.0, 127.7, 127.61, 127.56, 127.2, 126.8, 125.1, 124.0, 122.8, 118.9, 118.8, 108.9, 62.0, 14.1. IR (ATR, cm^−1^): 3219 (NH), 1682 (C=O), 1525, 1387, 1211, 756, 689. EI MS (70 eV): m/z (%): 496 (M+, 100), 467 (31), 234 (22). HRMS (ESI-QTOF) (M+H) calc. for C_32_H_24_N_4_O_2_: 497.1972 found: 497.1975.Ethyl 2-(3-((4-(4-chlorophenyl)pyrimidin-2-yl)amino)phenyl)quinoline-4-carboxylate (**14a**).From compounds **8** and **VIIIa**. Reaction time: 20 min. Yellow solid (81%) M.p. (214–216) °C. R_f_ Hex:AcOEt (7:3): 0.40. ^1^H NMR (400 MHz, DMSO-d_6_) δ 10.04 (s, 1H), 8.95 (s, 1H), 8.61 (d, *J* = 5.3 Hz, 1H), 8.59 (dd, *J* = 8.5, 1.0 Hz, 1H), 8.43 (s, 1H), 8.25 (d, *J* = 8.7 Hz, 2H), 8.15 (d, *J* = 8.4 Hz, 1H), 7.93–7.83 (m, 3H), 7.73 (ddd, *J* = 8.5, 7.0, 1.0 Hz, 1H), 7.56–7.50 (m, 3H), 7.48 (d, *J* = 5.3 Hz, 1H), 4.46 (q, *J* = 7.1 Hz, 2H), 1.37 (t, *J* = 7.1 Hz, 3H). ^13^C NMR (100 MHz, DMSO-d_6_) δ 165.8, 162.7, 159.8, 159.0, 156.0, 148.2, 141.1, 138.0, 136.6, 135.9, 135.4, 130.5, 129.7, 129.3, 129.0, 128.8, 128.1, 125.2, 123.3, 120.8, 120.7, 119.3, 117.7, 108.1, 62.0, 14.0. IR (ATR, cm^−1^): 3204 (NH), 1682 (C=O), 1559, 1191, 756. EI MS (70 eV): m/z (%): 480 (M+, 100), 451 (29), 369 (18), 226 (20), 203 (16). HRMS (ESI-QTOF) (M+H) calc. for C_28_H_21_ClN_4_O_2_: 481.1426 found: 481.1434.Ethyl 2-(3-((4-(naphthalen-2-yl)pyrimidin-2-yl)amino)phenyl)quinoline-4-carboxylate (**14b**).From compounds **8** and **VIIIb**. Reaction time: 20 min. Sonication helps the precipitation of **14b**. Yellow solid (78%) M.p. (139–141) °C. R_f_ Hex:AcOEt (7:3): 0.33. ^1^H NMR (400 MHz, DMSO-d_6_) δ 10.01 (s, 1H), 9.01 (s, 1H), 8.80 (s, 1H), 8.65 (d, *J* = 5.2 Hz, 1H), 8.61 (d, *J* = 8.5 Hz, 1H), 8.47 (s, 1H), 8.37 (d, *J* = 8.5 Hz, 1H), 8.15 (d, *J* = 8.4 Hz, 1H), 8.01–7.80 (m, 6H), 7.73 (pt, *J* = 7.7 Hz, 1H), 7.65–7.53 (m, 3H), 7.49 (pt, *J* = 7.5 Hz, 1H), 4.34 (q, *J* = 7.1 Hz, 2H), 1.28 (t, *J* = 7.1 Hz, 3H). ^13^C NMR (100 MHz, DMSO-d_6_) δ 165.8, 163.6, 160.1, 159.1, 156.1, 148.4, 141.3, 138.2, 136.4, 134.2, 134.0, 132.7, 130.4, 129.9, 129.3, 128.9, 128.5, 128.1, 127.6, 127.5, 127.2, 126.7, 125.2, 123.9, 123.3, 120.7, 120.5, 119.3, 117.7, 108.5, 61.8, 13.9. IR (ATR, cm^−1^): 3210 (NH), 1697 (C=O), 1536, 1513, 1391, 1223, 1173, 756. EI MS (70 eV): m/z (%): 496 (M+, 100), 467 (25), 234 (28), 211 (22), 44 (28). HRMS (ESI-QTOF) (M+H) calc. for C_32_H_24_N_4_O_2_: 497.1972 found: 497.1967.Ethyl 2-(2-((4-(4-chlorophenyl)pyrimidin-2-yl)amino)phenyl)quinoline-4-carboxylate (**15a**).From compounds **9** and **VIIIa**. Reaction time: 40 min. Purified by FCC using Hex:AcOEt (85:15). Yellow solid (78%) M.p. (118–120) °C. R_f_ Hex:AcOEt (8:2): 0.22. ^1^H NMR (400 MHz, CDCl_3_) δ 12.57 (s, 1H), 8.79 (d, *J* = 8.4 Hz, 1H), 8.76 (d, *J* = 8.5 Hz, 1H), 8.49 (d, *J* = 5.2 Hz, 1H), 8.44–8.38 (m, 2H), 8.05 (d, *J* = 8.6 Hz, 2H), 7.91 (dd, *J* = 8.1, 1.2 Hz, 1H), 7.85 (ddd, *J* = 8.4, 7.0, 1.2 Hz, 1H), 7.67 (ddd, *J* = 8.5, 7.0, 1.2 Hz, 1H), 7.53 (ddd, *J* = 8.4, 7.2, 1.2 Hz, 1H), 7.47 (d, *J* = 8.6 Hz, 2H), 7.20 (ddd, *J* = 8.1, 7.2, 1.2 Hz, 1H), 7.11 (d, *J* = 5.2 Hz, 1H), 4.55 (q, *J* = 7.1 Hz, 2H), 1.50 (t, *J* = 7.1 Hz, 3H). ^13^C NMR (100 MHz, CDCl_3_) δ 166.3, 163.6, 160.2, 158.5, 157.7, 147.5, 139.8, 137.1, 136.6, 135.6, 130.50, 130.47, 129.8, 129.7, 129.2, 128.5, 128.2, 125.5, 124.7, 123.4, 122.3, 122.0, 121.2, 108.1, 62.2, 14.5. IR (ATR, cm^−1^): (3300–2400) (wide NH signal), 1690 (C=O), 1501, 1410, 747. EI MS (70 eV): m/z (%): 480 (M+, 74), 407 (100), 280 (83). HRMS (ESI-QTOF) (M+H) calc. for C_28_H_21_ClN_4_O_2_: 481.1426 found: 481.1430.Ethyl 2-(2-((4-(naphthalen-2-yl)pyrimidin-2-yl)amino)phenyl)quinoline-4-carboxylate (**15b**).From compounds **9** and **VIIIb**. Reaction time: 40 min. Purified by FCC using Hex:AcOEt (85:15). Yellow solid (83%) M.p. (142–144) °C. R_f_ Hex:AcOEt (85:15): 0.18. ^1^H NMR (400 MHz, CDCl_3_) δ 12.66 (s, 1H), 8.94 (pd, *J* = 8.4 Hz, 1H), 8.78 (pd, *J* = 8.5 Hz, 1H), 8.64 (s, 1H), 8.55 (d, *J* = 5.2 Hz, 1H), 8.48–8.43 (m, 2H), 8.24 (dd, *J* = 8.5, 1.8 Hz, 1H), 8.00–7.85 (m, 5H), 7.69 (ddd, *J* = 8.5, 7.0, 1.2 Hz, 1H), 7.60–7.53 (m, 3H), 7.30 (d, *J* = 5.2 Hz, 1H), 7.20 (pt, *J* = 7.5 Hz, 1H), 4.55 (q, *J* = 7.1 Hz, 2H), 1.50 (t, *J* = 7.1 Hz, 3H). ^13^C NMR (100 MHz, CDCl_3_) δ 166.3, 164.5, 160.6, 158.7, 157.9, 147.5, 140.3, 136.6, 134.7, 134.6, 133.3, 130.6, 130.4, 129.8, 129.7, 129.1, 128.6, 128.2, 127.9, 127.44, 127.36, 126.6, 125.6, 124.4, 124.2, 123.5, 122.4, 121.6, 121.0, 108.7, 62.2, 14.5. IR (ATR, cm^−1^): (3200–2400) (wide NH signal), 1683 (C=O), 1508, 1406, 748. EI MS (70 eV): m/z (%): 496 (M+, 70), 423 (83), 296 (100). HRMS (ESI-QTOF) (M+H) calc. for C_32_H_24_N_4_O_2_: 497.1972 found: 497.1978.Ethyl 2-(2-((4-phenylpyrimidin-2-yl)amino)phenyl)quinoline-4-carboxylate (**15c**).From compounds **9** and **VIIIc**. Reaction time: 30 min. Purified by FCC using Hex:AcOEt (9:1). Yellow solid (85%) M.p. (95–97) °C. R_f_ Hex:AcOEt (8:2): 0.22. ^1^H NMR (400 MHz, CDCl_3_) δ 12.53 (s, 1H), 8.89 (dd, *J* = 8.5, 1.2 Hz, 1H), 8.77 (pd, *J* = 8.5 Hz, 1H), 8.51 (d, *J* = 5.2 Hz, 1H), 8.43 (s, 1H), 8.40 (pd, *J* = 8.4 Hz, 1H), 8.17–8.09 (m, 2H), 7.91 (dd, *J* = 7.9, 1.5 Hz, 1H), 7.85 (ddd, *J* = 8.4, 7.0, 1.3 Hz, 1H), 7.68 (ddd, *J* = 8.5, 7.0, 1.3 Hz, 1H), 7.59–7.49 (m, 4H), 7.23–7.14 (m, 2H), 4.55 (q, *J* = 7.1 Hz, 2H), 1.50 (t, *J* = 7.1 Hz, 3H). ^13^C NMR (100 MHz, CDCl_3_) δ 166.3, 164.6, 160.6, 158.7, 157.9, 147.6, 140.2, 137.4, 136.6, 130.8, 130.6, 130.4, 129.8, 129.7, 128.9, 128.2, 127.2, 125.5, 124.5, 123.4, 122.4, 121.6, 121.1, 108.5, 62.2, 14.5. IR (ATR, cm^−1^): (3200–2600) (wide NH signal), 1722 (C=O), 1529, 1432, 1273, 1192, 761. EI MS (70 eV): m/z (%): 446 (M+, 36), 373 (47), 246 (100), 186 (28). HRMS (ESI-QTOF) (M+H) calc. for C_28_H_22_N_4_O_2_: 447.1816 found: 447.1815.Ethyl (*E*)-2-(2-((4-styrylpyrimidin-2-yl)amino)phenyl)quinoline-4-carboxylate (**15d**).From compounds **9** and **VIIId**. Reaction temperature: 170 °C. Reaction time: 15 min. Purified by FCC using a gradient of Hex:AcOEt (9:1, 8:2). Yellow solid (65%) M.p. (120–122) °C. R_f_ Hex:AcOEt (8:2): 0.22. ^1^H NMR (400 MHz, CDCl_3_) δ 12.45 (s, 1H), 8.86 (d, *J* = 8.4 Hz, 1H), 8.77 (d, *J* = 8.5 Hz, 1H), 8.46–8.38 (m, 3H), 7.92–7.83 (m, 3H), 7.68 (ddd, *J* = 8.5, 7.0, 1.2 Hz, 1H), 7.63–7.59 (m, 2H), 7.54 (ddd, *J* = 8.4, 7.2, 1.2 Hz, 1H), 7.45–7.39 (m, 2H), 7.39–7.33 (m, 1H), 7.18 (ddd, *J* = 8.1, 7.2, 1.2 Hz, 1H), 7.00 (d, *J* = 15.9 Hz, 1H), 6.76 (d, *J* = 5.1 Hz, 1H), 4.55 (q, *J* = 7.1 Hz, 2H), 1.50 (t, *J* = 7.1 Hz, 3H). ^13^C NMR (100 MHz, CDCl_3_) δ 166.4, 163.0, 160.4, 158.6, 157.9, 147.6, 140.2, 136.6, 136.5, 136.1, 130.5, 130.4, 129.8, 129.7, 129.3, 129.0, 128.2, 127.7, 126.6, 125.6, 124.5, 123.4, 122.4, 121.6, 121.0, 110.7, 62.2, 14.5. IR (ATR, cm^−1^): (3200–2600) (wide NH signal), 2922, 2853, 1717 (C=O), 1529, 1432, 763. EI MS (70 eV): m/z (%): 472 (M+, 36), 399 (45), 272 (100), 128 (33). HRMS (ESI-QTOF) (M+H) calc. for C_30_H_24_N_4_O_2_: 473.1972 found: 473.1975.Ethyl 2-(2-((4-(4-chlorophenyl)pyrimidin-2-yl)amino)phenyl)-6-fluoroquinoline-4-carboxylate (**16a**).From compounds **10** and **VIIIa**. Reaction time: 30 min. Yellow solid (88%) M.p. (139–141) °C. R_f_ Hex:AcOEt (8:2): 0.16. ^1^H NMR (400 MHz, CDCl_3_) δ 12.35 (s, 1H), 8.77 (d, *J* = 8.4 Hz, 1H), 8.52 (dd, *^3^J_HF_* = 10.7 Hz, *J* = 2.8 Hz, 1H), 8.50–8.47 (m, 2H), 8.41 (dd, *J* = 9.2 Hz, *^4^J_HF_* = 5.6 Hz, 1H), 8.03 (d, *J* = 8.6 Hz, 2H), 7.88 (dd, *J* = 8.1, 1.3 Hz, 1H), 7.61 (ddd, *J* = 9.2, 2.8 Hz, *^3^J_HF_* = 8.0 Hz, 1H), 7.53 (ddd, *J* = 8.4, 7.2, 1.3 Hz, 1H), 7.47 (d, *J* = 8.6 Hz, 2H), 7.21 (pt, *J* = 7.5 Hz, 1H), 7.12 (d, *J* = 5.3 Hz, 1H), 4.54 (q, *J* = 7.1 Hz, 2H), 1.50 (t, *J* = 7.1 Hz, 3H). ^13^C NMR (100 MHz, CDCl_3_) δ 165.8, 163.8, 161.8 (d, *^1^J_CF_* = 249.1 Hz), 160.0, 158.3, 157.1 (d, *^6^J_CF_* = 2.9 Hz), 144.8, 139.6, 137.2, 135.7 (d, *^4^J_CF_* = 5.6 Hz), 135.6, 132.1 (d, *^3^J_CF_* = 9.3 Hz), 130.5, 129.8, 129.2, 128.5, 124.7, 124.4 (d, *^3^J_CF_* = 11.6 Hz), 123.4, 122.1, 121.3, 120.7 (d, *^2^J_CF_* = 26.2 Hz), 109.8 (d, *^2^J_CF_* = 25.2 Hz), 108.1, 62.3, 14.5. IR (ATR, cm^−1^): (3200–2500) (wide NH signal), 1713 (C=O), 1535, 1440, 1224, 806. EI MS (70 eV): m/z (%): 498 (M+, 35), 425 (30), 280 (100). HRMS (ESI-QTOF) (M+H) calc. for C_28_H_20_ClFN_4_O_2_: 499.1332 found: 499.1334.Ethyl 6-fluoro-2-(2-((4-(naphthalen-2-yl)pyrimidin-2-yl)amino)phenyl)quinoline-4-carboxylate (**16b**).From compounds **10** and **VIIIb**. Reaction time: 20 min. Yellow solid (89%) M.p. (141–143) °C. R_f_ Hex:AcOEt (8:2): 0.25. ^1^H NMR (400 MHz, CDCl_3_) δ 12.44 (s, 1H), 8.92 (dd, *J* = 8.4, 1.1 Hz, 1H), 8.61 (s, 1H), 8.57–8.49 (m, 3H), 8.47–8.40 (m, 1H), 8.22 (pd, *J* = 8.5 Hz, 1H), 8.00–7.93 (m, 2H), 7.93–7.87 (m, 2H), 7.68–7.60 (m, 1H), 7.60–7.53 (m, 3H), 7.34–7.27 (m, 1H), 7.21 (pt, *J* = 7.6 Hz, 1H), 4.54 (q, *J* = 7.1 Hz, 2H), 1.50 (t, *J* = 7.1 Hz, 3H). ^13^C NMR (100 MHz, CDCl_3_) δ 165.9, 164.6, 161.8 (d, *^1^J_CF_* = 249.0 Hz), 160.5, 158.6, 157.2, 144.8, 140.1, 135.6, 134.7, 134.6, 133.3, 132.0 (d, *^3^J_CF_* = 9.5 Hz), 130.6, 129.7, 129.1, 128.7, 127.9, 127.5, 127.4, 126.7, 124.4 (d, *^3^J_CF_* = 10.8 Hz), 124.3, 124.1, 123.4, 121.8, 121.2, 120.6 (d, *^2^J_CF_* = 25.8 Hz), 109.8 (d, *^2^J_CF_* = 25.3 Hz), 108.7, 62.3, 14.5. IR (ATR, cm^−1^): (3300–2600) (wide NH signal), 1718 (C=O), 1535, 1434, 1270, 1222, 808, 747. EI MS (70 eV): m/z (%): 514 (M+, 30), 441 (24), 296 (100). HRMS (ESI-QTOF) (M+H) calc. for C_32_H_23_FN_4_O_2_: 515.1878 found: 515.1879.Ethyl 6-fluoro-2-(2-((4-phenylpyrimidin-2-yl)amino)phenyl)quinoline-4-carboxylate (**16c**).From compounds **10** and **VIIIc**. Reaction time: 20 min. Yellow solid (84%) M.p. (122–124) °C. R_f_ Hex:AcOEt (8:2): 0.24. ^1^H NMR (400 MHz, CDCl_3_) δ 12.35 (s, 1H), 8.87 (pd, *J* = 8.4 Hz, 1H), 8.53 (dd, *^3^J_HF_* = 10.7 Hz, *J* = 2.8 Hz, 1H), 8.51–8.48 (m, 2H), 8.38 (dd, *J* = 9.2 Hz, *^4^J_HF_* = 5.6 Hz, 1H), 8.14–8.09 (m, 2H), 7.88 (dd, *J* = 7.9, 1.4 Hz, 1H), 7.61 (ddd, *J* = 9.2, 2.8 Hz, *^3^J_HF_* = 8.0 Hz, 1H), 7.57–7.49 (m, 4H), 7.23–7.13 (m, 2H), 4.54 (q, *J* = 7.1 Hz, 2H), 1.50 (t, *J* = 7.1 Hz, 3H). ^13^C NMR (100 MHz, CDCl_3_) δ 165.9, 164.7, 161.7 (d, *^1^J_CF_* = 249.2 Hz), 160.5, 158.7, 157.2 (d, *^6^J_CF_* = 2.8 Hz), 144.8, 140.1, 137.3, 135.5 (d, *^4^J_CF_* = 6.1 Hz), 132.0 (d, *^3^J_CF_* = 9.2 Hz), 130.8, 130.6, 129.7, 128.9, 127.2, 124.4 (d, *^3^J_CF_* = 11.0 Hz), 124.2, 123.4, 121.7, 121.1, 120.6 (d, *^2^J_CF_* = 26.1 Hz), 109.8 (d, *^2^J_CF_* = 25.0 Hz), 108.5, 62.3, 14.4. IR (ATR, cm^−1^): (3200–2500) (wide NH signal), 1721 (C=O), 1537, 1436, 1271, 1223, 767. EI MS (70 eV): m/z (%): 464 (M+, 33), 391 (28), 246 (100). HRMS (ESI-QTOF) (M+H) calc. for C_28_H_21_FN_4_O_2_: 465.1721 found: 465.1724.Ethyl (*E*)-6-fluoro-2-(2-((4-styrylpyrimidin-2-yl)amino)phenyl)quinoline-4-carboxylate (**16d**).From compounds **10** and **VIIId**. Reaction temperature: 170 °C. Reaction time: 10 min. Yellow solid (70%) M.p. (147–149) °C. R_f_ Hex:AcOEt (8:2): 0.24. ^1^H NMR (400 MHz, CDCl_3_) δ 12.23 (s, 1H), 8.84 (dd, *J* = 8.4, 1.1 Hz, 1H), 8.53 (dd, *^3^J_HF_* = 10.7 Hz, *J* = 2.8 Hz, 1H), 8.49 (s, 1H), 8.42 (d, *J* = 5.1 Hz, 1H), 8.39 (dd, *J* = 9.2 Hz, *^4^J_HF_* = 5.6 Hz, 1H), 7.91–7.80 (m, 2H), 7.65–7.58 (m, 3H), 7.53 (ddd, *J* = 8.4, 7.2, 1.5 Hz, 1H), 7.45–7.39 (m, 2H), 7.39–7.33 (m, 1H), 7.18 (ddd, *J* = 8.1, 7.2, 1.2 Hz, 1H), 6.99 (d, *J* = 15.9 Hz, 1H), 6.76 (d, *J* = 5.1 Hz, 1H), 4.54 (q, *J* = 7.1 Hz, 2H), 1.50 (t, *J* = 7.1 Hz, 4H). ^13^C NMR (100 MHz, CDCl_3_) δ 165.9, 163.0, 161.8 (d, *^1^J_CF_* = 250.4 Hz), 160.3, 158.6, 157.3 (d, *^6^J_CF_* = 2.8 Hz), 144.8, 140.1, 136.5, 136.1, 135.6 (d, *^4^J_CF_* = 5.7 Hz), 132.0 (d, *^3^J_CF_* = 9.7 Hz), 130.6, 129.7, 129.3, 129.0, 127.7, 126.6, 124.4 (d, *^3^J_CF_* = 11.0 Hz), 124.3, 123.5, 121.7, 121.1, 120.6 (d, *^2^J_CF_* = 25.8 Hz), 110.8, 109.8 (d, *^2^J_CF_* = 25,3 Hz), 62.3, 14.5. IR (ATR, cm^−1^): (3200–2500) (wide NH signal), 2922, 2853, 1713 (C=O), 1520, 1402, 1338, 1278, 1230, 744, 699. EI MS (70 eV): m/z (%): 490 (M+, 35), 417 (18), 272 (100). HRMS (ESI-QTOF) (M+H) calc. for C_30_H_23_FN_4_O_2_: 491.1878 found: 491.1881.Crystals suitable for X-ray single-crystal diffraction were obtained from DMSO solution, and the crystal data for **16d** were deposited at CCDC with reference CCDC 2368196: Chemical formula C_30_H_23_FN_4_O_2_, Mr 490.52; Monoclinic, P21/n; 100 K, Cell dimensions a, b, c (Å) 12.2248(7), 7.8728(5), 24.2984(16) β (°) α, β, γ (°) 90, 99.681(2), 90. V (Å^3^) 2305.3(2), Z = 4, F (000) = 1024, Dx (Mg m^−3^) = 1.413, Mo Kα, μ (mm^−1^) = 0.096, Crystal size (mm) = 0.24 × 0.09 × 0.08. Data collection: Diffractometer Bruker D8 Venture (APEX 3), monochromator multilayer mirror, CCD rotation images, thick slices φ and θ scans, Mo INCOATEC high-brilliance microfocus sealed tube (λ = 0.71073 Å), multiscan absorption correction (SADABS 2016/2), Tmin, Tmax 0.660, 0.746. No. of measured, independent and observed [I > 2σ(I)] reflections 54,633, 5286, 4316, Rint = 0. 087, (sin θ/λ)max (Å^−1^) 0.411, θ values (°): θmax = 27.5, θmin = 2.0; Range h = −15→15, k = −10→10, l = −31→31, Refinement on F^2^:R[F^2^ > 2σ(F^2^)] = 0. 063, wR(F^2^) = 0.166, S=1.079. No. of reflections 5286, No. of parameters 339, No. of restraints 0. Weighting scheme: w = 1/σ^2^(Fo^2^) + (0.0753P)^2^ + 2.8379P where P = (Fo^2^ + 2Fc^2^)/3. (∆/σ) < 0.012, Δρmax, Δρmin (e Å^−3^) 0.347, −0.31. Initial R was higher than 0.10, and after running TwinRotMat using PLATON (version 260918), a new hkl file was generated as hklf5 and so refined as two-component nonmerohedral twinning related by (0.689, 0.000, −0.311; 0.000, −1.0000, 0.000; −1.689, 0.000, −0.689) matrix with BASF 0.24.Ethyl 6-chloro-2-(2-((4-(4-chlorophenyl)pyrimidin-2-yl)amino)phenyl)quinoline-4-carboxylate (**17a**).From compounds **11** and **VIIIa**. Reaction time: 20 min. Orange solid (85%) M.p. (140–142) °C. R_f_ Hex:AcOEt (8:2): 0.28. ^1^H NMR (400 MHz, CDCl_3_) δ 12.41 (s, 1H), 8.84 (d, *J* = 2.3 Hz, 1H), 8.81 (dd, *J* = 8.4, 1.2 Hz, 1H), 8.49 (d, *J* = 5.2 Hz, 1H), 8.45 (s, 1H), 8.28 (d, *J* = 8.9 Hz, 1H), 8.03 (d, *J* = 8.7 Hz, 2H), 7.88 (dd, *J* = 8.1, 1.3 Hz, 1H), 7.76 (dd, *J* = 8.9, 2.3 Hz, 1H), 7.53 (ddd, *J* = 8.4, 7.2, 1.3 Hz, 1H), 7.47 (d, *J* = 8.7 Hz, 2H), 7.18 (ddd, *J* = 8.1, 7.2, 1.2 Hz, 1H), 7.10 (d, *J* = 5.2 Hz, 1H), 4.54 (q, *J* = 7.1 Hz, 2H), 1.50 (t, *J* = 7.1 Hz, 3H). ^13^C NMR (100 MHz, CDCl_3_) δ 165.7, 163.5, 160.4, 158.8, 158.0, 145.9, 140.0, 137.0, 135.7, 135.3, 134.4, 131.4, 131.0, 130.8, 129.8, 129.2, 128.4, 124.8, 124.1, 124.0, 123.3, 121.8, 121.1, 108.2, 62.4, 14.5. IR (ATR, cm^−1^): (3200–2700) (wide NH signal), 1715 (C=O), 1535, 1435, 1269, 1178, 803. EI MS (70 eV): m/z (%): 514 (M+, 27), 441 (38), 280 (100). HRMS (ESI-QTOF) (M+H) calc. for C_28_H_20_Cl_2_N_4_O_2_: 515.1036 found: 515.1037.Ethyl 6-chloro-2-(2-((4-(naphthalen-2-yl)pyrimidin-2-yl)amino)phenyl)quinoline-4-carboxylate (**17b**).From compounds **11** and **VIIIb**. Reaction time: 20 min. Yellow solid (91%) M.p. (165–167) °C. R_f_ Hex:AcOEt (8:2): 0.25. ^1^H NMR (400 MHz, CDCl_3_) δ 12.54 (s, 1H), 8.93 (dd, *J* = 8.4, 1.3 Hz, 1H), 8.85 (d, *J* = 2.3 Hz, 1H), 8.60 (s, 1H), 8.53 (d, *J* = 5.2 Hz, 1H), 8.47 (s, 1H), 8.33 (d, *J* = 8.9 Hz, 1H), 8.20 (dd, *J* = 8.6, 1.8 Hz, 1H), 7.98–7.93 (m, 2H), 7.91–7.87 (m, 2H), 7.78 (dd, *J* = 8.9, 2.3 Hz, 1H), 7.59–7.53 (m, 3H), 7.28 (d, *J* = 5.2 Hz, 1H), 7.18 (ddd, *J* = 8.1, 7.2, 1.3 Hz, 1H), 4.54 (q, *J* = 7.1 Hz, 2H), 1.50 (t, *J* = 7.1 Hz, 3H). ^13^C NMR (100 MHz, CDCl_3_) δ 165.7, 164.5, 160.5, 158.7, 158.0, 145.9, 140.4, 135.2, 134.7, 134.5, 134.3, 133.3, 131.3, 131.0, 130.8, 129.7, 129.1, 128.6, 127.9, 127.4, 126.6, 124.8, 124.1, 124.0, 123.8, 123.3, 121.6, 121.0, 108.7, 62.3, 14.5. IR (ATR, cm^−1^): (3200–2600) (wide NH signal), 2921, 2852, 1721 (C=O), 1535, 1429, 1270, 1183, 808, 744. EI MS (70 eV): m/z (%): 530 (M+, 29), 457 (63), 296 (100). HRMS (ESI-QTOF) (M+H) calc. for C_32_H_23_ClN_4_O_2_: 531.1582 found: 531.1580.Ethyl 6-chloro-2-(2-((4-phenylpyrimidin-2-yl)amino)phenyl)quinoline-4-carboxylate (**17c**).From compounds **11** and **VIIIc**. Reaction time: 20 min. Yellow solid (80%) M.p. (170–172) °C. R_f_ Hex:AcOEt (8:2): 0.29. ^1^H NMR (400 MHz, CDCl_3_) δ 12.40 (s, 1H), 8.90–8.83 (m, 2H), 8.50 (d, *J* = 5.2 Hz, 1H), 8.47 (s, 1H), 8.30 (d, *J* = 8.9 Hz, 1H), 8.14–8.08 (m, 2H), 7.88 (dd, *J* = 8.1, 1.2 Hz, 1H), 7.77 (dd, *J* = 8.9, 2.3 Hz, 1H), 7.57–7.48 (m, 4H), 7.21–7.14 (m, 2H), 4.54 (q, *J* = 7.1 Hz, 2H), 1.50 (t, *J* = 7.1 Hz, 3H). ^13^C NMR (100 MHz, CDCl_3_) δ 165.7, 164.7, 160.5, 158.7, 158.1, 146.0, 140.2, 137.3, 135.3, 134.3, 131.4, 131.0, 130.84, 130.80, 129.7, 128.9, 127.2, 124.8, 124.04, 124.02, 123.4, 121.6, 121.1, 108.6, 62.3, 14.5. IR (ATR, cm^−1^): (3200–2600) (wide NH signal), 1719 (C=O), 1536, 1430, 1334, 1269, 764. EI MS (70 eV): m/z (%): 480 (M+, 28), 407 (60), 246 (100). HRMS (ESI-QTOF) (M+H) calc. for C_28_H_21_ClN_4_O_2_: 481.1426 found: 481.1428.Ethyl (*E*)-6-chloro-2-(2-((4-styrylpyrimidin-2-yl)amino)phenyl)quinoline-4-carboxylate (**17d**).From compounds **11** and **VIIId**. Reaction temperature: 170 °C. Reaction time: 10 min. Orange solid (72%) M.p. (166–168) °C. R_f_ Hex:AcOEt (8:2): 0.27. ^1^H NMR (400 MHz, CDCl_3_) δ 12.30 (s, 1H), 8.90–8.81 (m, 2H), 8.46 (s, 1H), 8.42 (d, *J* = 5.1 Hz, 1H), 8.31 (dd, *J* = 8.9, 0.4 Hz, 1H), 7.89–7.86 (m, 1H), 7.84 (d, *J* = 16.3 Hz, 2H), 7.77 (dd, *J* = 8.9, 2.3 Hz, 1H), 7.62–7.58 (m, 2H), 7.53 (ddd, *J* = 8.4, 7.2, 1.2 Hz, 1H), 7.45–7.39 (m, 2H), 7.38–7.33 (m, 1H), 7.17 (ddd, *J* = 8.1, 7.2, 1.2 Hz, 1H), 6.98 (d, *J* = 15.9 Hz, 1H), 6.75 (d, *J* = 5.1 Hz, 1H), 4.54 (q, *J* = 7.1 Hz, 2H), 1.50 (t, *J* = 7.1 Hz, 3H). ^13^C NMR (100 MHz, CDCl_3_) δ 165.7, 163.0, 160.3, 158.6, 158.1, 145.9, 140.2, 136.5, 136.0, 135.3, 134.3, 131.3, 131.1, 130.7, 129.7, 129.3, 129.0, 127.7, 126.6, 124.8, 124.04, 124.00, 123.3, 121.6, 121.0, 110.8, 62.3, 14.5. IR (ATR, cm^−1^): (3200–2600) (wide NH signal), 2922, 2852, 1719 (C=O), 1534, 1446, 1432, 1372, 1270, 1180, 746. EI MS (70 eV): m/z (%): 506 (M+, 25), 433 (38), 272 (100), 87 (48). HRMS (ESI-QTOF) (M+H) calc. for C_30_H_23_ClN_4_O_2_: 507.1582 found: 507.1587.Ethyl 6-bromo-2-(2-((4-(4-chlorophenyl)pyrimidin-2-yl)amino)phenyl)quinoline-4-carboxylate (**18a**).From compounds **12** and **VIIIa**. Reaction time: 20 min. Purified by FCC using Hex:AcOEt (8:2). Yellow solid (70%) M.p. (161–163) °C. R_f_ Hex:AcOEt (8:2): 0.22. ^1^H NMR (400 MHz, CDCl_3_) δ 12.43 (s, 1H), 9.01 (d, *J* = 2.1 Hz, 1H), 8.82 (dd, *J* = 8.4, 1.3 Hz, 1H), 8.49 (d, *J* = 5.2 Hz, 1H), 8.45 (s, 1H), 8.20 (d, *J* = 8.9 Hz, 1H), 8.03 (d, *J* = 8.6 Hz, 2H), 7.92–7.86 (m, 2H), 7.53 (ddd, *J* = 8.4, 7.2, 1.3 Hz, 1H), 7.47 (d, *J* = 8.6 Hz, 2H), 7.18 (ddd, *J* = 8.1, 7.2, 1.3 Hz, 1H, 1H), 7.10 (d, *J* = 5.2 Hz, 1H), 4.54 (q, *J* = 7.1 Hz, 2H), 1.50 (t, *J* = 7.1 Hz, 3H). ^13^C NMR (100 MHz, CDCl_3_) δ 165.7, 163.4, 160.5, 158.9, 158.1, 146.1, 140.1, 137.0, 135.7, 135.2, 134.0, 131.0, 130.8, 129.8, 129.2, 128.4, 128.1, 124.5, 124.0, 123.3, 122.7, 121.8, 121.0, 108.2, 62.4, 14.5. IR (ATR, cm^−1^): 3273 (NH), 2922, 2852, 1725 (C=O), 1532, 1415, 1268, 1176, 803, 743. EI MS (70 eV): m/z (%): 558 (M+, 21), 485 (29), 280 (100). HRMS (ESI-QTOF) (M+H) calc. for C_28_H_20_BrClN_4_O_2_: 559.0531 found: 559.0530.Ethyl 6-bromo-2-(2-((4-(naphthalen-2-yl)pyrimidin-2-yl)amino)phenyl)quinoline-4-carboxylate (**18b**).From compounds **12** and **VIIIb**. Reaction time: 20 min. Yellow solid (84%) M.p. (170–172) °C. R_f_ Hex:AcOEt (8:2): 0.22. ^1^H NMR (400 MHz, CDCl_3_) δ 12.53 (s, 1H), 9.01 (d, *J* = 2.1 Hz, 1H), 8.92 (d, *J* = 8.4 Hz, 1H), 8.59 (s, 1H), 8.52 (d, *J* = 5.2 Hz, 1H), 8.46 (s, 1H), 8.26 (d, *J* = 8.9 Hz, 1H), 8.20 (dd, *J* = 8.6, 1.8 Hz, 1H), 7.99–7.92 (m, 2H), 7.91–7.87 (m, 3H), 7.58–7.53 (m, 3H), 7.29 (d, *J* = 5.2 Hz, 1H), 7.18 (ddd, *J* = 8.1, 7.2, 1.3 Hz, 1H), 4.54 (q, *J* = 7.1 Hz, 2H), 1.50 (t, *J* = 7.1 Hz, 3H). ^13^C NMR (100 MHz, CDCl_3_) δ 165.7, 164.5, 160.5, 158.6, 158.2, 146.1, 140.3, 135.1, 134.7, 134.5, 133.9, 133.3, 131.1, 130.9, 129.7, 129.1, 128.6, 128.1, 127.9, 127.4, 126.7, 124.5, 124.1, 123.9, 123.2, 122.7, 121.6, 121.1, 108.7, 62.3, 14.4. IR (ATR, cm^−1^): (3200–2600) (wide NH signal), 1720 (C=O), 1534, 1427, 1269, 807, 741. EI MS (70 eV): m/z (%): 574 (M+, 20), 501 (19), 296 (100), 152 (29). HRMS (ESI-QTOF) (M+H) calc. for C_32_H_23_BrN_4_O_2_: 575.1077 found: 575.1075.Ethyl 6-bromo-2-(2-((4-phenylpyrimidin-2-yl)amino)phenyl)quinoline-4-carboxylate (**18c**).From compounds **12** and **VIIIc**. Reaction time: 20 min. Yellow solid (76%) M.p. (171–173) °C. R_f_ Hex:AcOEt (8:2): 0.29. ^1^H NMR (400 MHz, CDCl_3_) δ 12.39 (s, 1H), 9.02 (d, *J* = 2.1 Hz, 1H), 8.88 (dd, *J* = 8.4, 1.2 Hz, 1H), 8.50 (d, *J* = 5.2 Hz, 1H), 8.46 (s, 1H), 8.24 (d, *J* = 8.9 Hz, 1H), 8.14–8.09 (m, 2H), 7.93–7.87 (m, 2H), 7.58–7.48 (m, 4H), 7.23–7.14 (m, 2H), 4.54 (q, *J* = 7.1 Hz, 2H), 1.50 (t, *J* = 7.1 Hz, 3H). ^13^C NMR (100 MHz, CDCl_3_) δ 165.7, 164.7, 160.5, 158.7, 158.2, 146.2, 140.2, 137.3, 135.2, 134.0, 131.1, 130.86, 130.84, 129.8, 128.9, 128.1, 127.2, 124.5, 124.1, 123.3, 122.7, 121.7, 121.1, 108.6, 62.4, 14.5. IR (ATR, cm^−1^): (3200–2700) (wide NH signal), 1720 (C=O), 1535, 1429, 1270, 1180, 764. EI MS (70 eV): m/z (%): 524 (M+, 18), 451 (20), 246 (100), 186 (25). HRMS (ESI-QTOF) (M+H) calc. for C_28_H_21_BrN_4_O_2_: 525.0921 found: 525.0918.Ethyl (*E*)-6-bromo-2-(2-((4-styrylpyrimidin-2-yl)amino)phenyl)quinoline-4-carboxylate (**18d**).From compounds **12** and **VIIId**. Reaction time: 20 min. Yellow solid (75%) M.p. (198–200) °C. R_f_ Hex:AcOEt (8:2): 0.28. ^1^H NMR (400 MHz, CDCl_3_) δ 12.29 (s, 1H), 9.02 (d, *J* = 2.1 Hz, 1H), 8.84 (d, *J* = 8.4 Hz, 1H), 8.45 (s, 1H), 8.42 (d, *J* = 5.1 Hz, 1H), 8.23 (d, *J* = 8.9 Hz, 1H), 7.92–7.86 (m, 2H), 7.84 (d, *J* = 15.9 Hz, 1H), 7.60 (d, *J* = 7.1 Hz, 2H), 7.54 (pt, *J* = 7.6 Hz, 1H), 7.45–7.39 (m, 2H), 7.39–7.33 (m, 1H), 7.17 (pt, *J* = 7.6 Hz, 1H), 6.98 (d, *J* = 15.9 Hz, 1H), 6.76 (d, *J* = 5.1 Hz, 1H), 4.54 (q, *J* = 7.1 Hz, 2H), 1.50 (t, *J* = 7.1 Hz, 3H). ^13^C NMR (100 MHz, CDCl_3_) δ 165.7, 163.0, 160.2, 158.6, 158.2, 146.1, 140.2, 136.5, 136.0, 135.2, 133.9, 131.1, 130.8, 129.8, 129.3, 129.0, 128.1, 127.7, 126.5, 124.5, 124.0, 123.3, 122.6, 121.6, 121.0, 110.8, 62.3, 14.5. IR (ATR, cm^−1^): (3200–2700) (wide NH signal), 1720 (C=O), 1534, 1445, 1270, 745. EI MS (70 eV): m/z (%): 550 (M+, 21), 477 (18), 272 (100), 128 (18), 127 (21). HRMS (ESI-QTOF) (M+H) calc. for C_30_H_23_BrN_4_O_2_: 551.1077 found: 551.1079.

### 3.3. Molecular Modelling

The molecular modelling and docking analysis were performed using the MOE 2022 suit from Chemical Computing Group’s Molecular Operating Environment, and the minimization of the energy of molecules and complexes was performed under molecular mechanics using the Amber14:EHT force field.

The complexes of the hLDHA protein with the inhibitor **W31** (PDB code **4R68**) and hLDHB protein with the inhibitor **Oxamate** (PDB code **1I0Z**) and Oxamate-**H1U** (PDB code **7DBJ**) were downloaded from the Protein Data Bank (PDB) and prepared as follows: For **4R68**, **1I0Z**, and **7DBJ** at the **OXM** site, all the chains along with their respective ligands and solvents, save for chain A, were deleted using the sequence editor (SEQ); for 7DBJ at the HIU site, which is located between two chains (A–C and B–D), chains B–D along with their respective ligands and solvents were deleted using the sequence editor (SEQ). Then, quick preparation was completed, performing the structure preparation for docking, including the following steps: (i) fixing the issues in the protein structure. There were structural problems that needed to be addressed before proceeding with a simulation, and most items were identified and corrected automatically by the application. In all of these instances, no additional actions needed to be taken. (ii) Protonation, which also analysed residues in which it was possible to have multiple protomers and/or a tautomer and checked for the right charge in any heteroatom, ultimately minimizing the complex system using the force field Amber14. (iii) It was confirmed that no warnings that were not automatically corrected and the user should be aware of were displayed. Then, the complexes were submitted to the general energy minimization mode, in which force field minimization was performed. No restraints were applied, and tethers of different strengths were set on the receptor, ligand, and solvent atoms. Constraints were selected in order to maintain rigid water molecules. The gradient was of 0.1 RMS, meaning that the energy minimization was finished when the root mean square gradient fell below the specified value (0.1).

The input database of screened molecules was prepared from builder editor and imported in the corresponding database file (*.mdb), which was used as the input file in the docking process. To prepare the database input file, we followed a similar preparation process that included a first wash (a set of cleaning rules to ensure that each structure is in a suitable form for subsequent modelling steps, such as conformational enumeration and protein-ligand docking), checking for the right partial charges, and finally, minimizing the energy of the molecules using the force field Amber14.

For hLDHA docking, three pharmacophoric models were created from the Pharmacophore Query Editor tool: (i) **W31** site, (ii) NADH site, and (iii) extended site **W31**-NADH site. Three features were defined so as to interact with the main amino acid residues: Arg^168^, His^192^, Asn^137^, and Asp^194^. All three features were defined with a radius of 1.2 Å, and none of them were classified as essential nor ignored. When establishing the search criteria, the partial match was clicked on and defined as at least one interaction with one of those features. The docking screening was carried out with the following settings: (i) receptor, MOE (the previously prepared complex) receptor atoms; (ii) site, ligand atoms, (iii) pharmacophore, on; (iv) conformation import for analysis of ligand conformation; (v) placement, pharmacophore; (vi) number of returned poses (poses returned by each ligand’s placement), 3000; (vii) placement score, London dG; (viii) placement poses, 100; (ix) refinement method, rigid receptor; (x) refinement score, GBI/WSA dG; refinement poses (number of poses retained to be written in the output file), 10.

Once the docking was complete, the best pose score for each ligand determined by a further minimization process (in the output file) was required using molecular mechanics and the specified forcefield. The best pose was determined by the following criteria: (i) RMSD < 1.8 Å [99], (ii) affinity (S) [100] values < −9 kcal/mol, and (iii) energy values involved in the interactions with the main amino acid residues [100], selecting those interacting with Arg168 first, followed by those with a higher number of interactions. In the case that they all interacted with the same amino acids, the ones with the highest energy values involved in the interactions with those amino acid residues were chosen.

For hLDHB docking, no pharmacophoric models were created neither for the OXM nor the extended (**OXM**-NADH) or the **H1U** sites. The docking screening was carried out with the following settings: (i) receptor, MOE (the previously prepared complex), receptor atoms; (ii) site, ligand atoms; (iii) conformation import for analysis of ligand conformations; (iv) placement: triangle matcher; (v) number of returned poses (poses returned by each ligand’s placement), 3000; (vi) placement score, London dG; placement poses, 100; (vii) refinement method, rigid receptor; (viii) refinement score, GBI/WSA dG; (ix) refinement poses (number of poses retained to be written in the output file), 10.

For the initial screening, the best pose was determined, in both complexes, according to the best raw affinity value. For the in-depth analysis, the best pose was determined as follows: Those poses with an RMSD < 1.8 Å were selected [99] for further minimization using molecular mechanics and the specified forcefield. Then, the best pose was defined according to the best (or higher number of) interactions with the amino acid residues [98] that interacted with reference NADH (in case of complex **1I0Z**) or with Glu^214^ (in complex **7DBJ**).

### 3.4. Human Lactate Dehydrogenase a Enzymatic Activity Assay

The ability of the synthesized compounds to inhibit hLDHA enzyme was measured using recombinant human LDHA (95%, specific activity > 300 units/mg, and concentration 0.5 mg/mL, Abcam, Cambridge, UK) with sodium pyruvate (96%, Merck) as substrate and β-NADH (≥97%, Merck) as cofactor in potassium phosphate buffer (100 mM, pH 7.3). The enzymatic assay was conducted on 96-well microplates, and the decrease in the β-NADH fluorescence (λ_excitation_ = 340 nm; λ_emission_ = 460 nm) was detected in a TECAN Infinite 200 Pro M Plex fluorescent plate reader (Tecan Instrument, Inc., Männedorf, Switzerland) at 28 °C. The activity was determined according to our previously described protocol [44,46]. The final volume in each well was set to 200 µL using 100 mM potassium phosphate buffer, 0.041 units/mL LDHA, 155 µM β-NADH, 1 mM pyruvate (saturated conditions), and DMSO solutions (5%, *v/v*) of pure compounds at concentrations in the range of 0.048 to100 µM. The addition of pyruvate enabled the beginning of the reaction, and fluorescence was registered every 60 s over the course of 10 min. A lineal time interval was selected to calculate the slope at every single concentration. The establishment of the 0% and 100% enzymatic activity was performed by controls and by the use of the inhibitor 3-[[3[(cyclopropylamino)sulfonyl]-7-(2,4-dimethoxy-5-pyrimidinyl)-4-quinolinyl]amino]-5-(3,5-difluorophenoxy)benzoic acid (GSK 2837808 A, Tocris, Minneapolis, MN, USA) at 1 µM [41]. The slope obtained at each concentration was compared to the one obtained for the 100% enzymatic activity control to determine the corresponding enzymatic activity. A nonlinear regression analysis in GraphPad Prism version 9.00 for Windows (GraphPad Software, La Jolla, CA, USA) was used for the dose-response curve, fitting of the logarithm of inhibitor concentration vs. normalized enzymatic activity, and to calculate IC_50_ values (see Appendix A). All measurements were taken in triplicate, and data were expressed as the mean ± SD.

### 3.5. Human Lactate Dehydrogenase B Enzymatic Activity Assay

The ability of the synthesized compounds to inhibit hLDHB enzyme was determined using recombinant human LDHB (95%, specific activity > 300 units/mg, and concentration 1.0 mg/mL, Abcam, Cambridge, UK) following the same fluorimetric protocol described in Section 3.4.

## 4. Conclusions

We have designed and synthesised a new family of ethyl pyrimidine-quinolinecarboxylate derivatives based on our previous pyrimidine-quinolone hybrids, which showed moderate inhibition of hLDHA. Our primary modification to the quinoline moiety consisted of adding a carboxylic group, which acted as a good hydrogen bonding acceptor/donor fragment. This new family successfully passed virtual docking screening and was synthesized using the Pfitzinger reaction to produce key ethyl 2-aminophenylquinoline-4-carboxylate intermediates (**7**–**12**). These intermediates were then coupled with a set of 4-aryl-2-chloropyrimidines **VIII**(**a**–**d**) using microwave irradiation at 150–170 °C and ethanol as a green solvent, affording the ethyl pyrimidine-quinolinecarboxylate derivatives in moderate to good yields.

The hLDHA inhibition assay results agreed with the docking, showing that the most potent compounds were those with a 2-aminophenyl group linking both quinoline and pyrimidine scaffolds in a U-shaped arrangement that fit well in the enzyme’s active site.

As a result, we identified thirteen compounds with IC_50_ values of less than 5 μM, and four of them (**16a**, **18b**, **18c**, and **18d**) with IC_50_ values of approximately 1 μM. This represents an average of a 20-fold improvement in their inhibitory activity compared to the result obtained with the previous family of compounds.

In addition, this study evaluated all hybrids with IC_50_ < 10 μM against the hLDHB isoenzyme. Three of the hybrids (**15c**, **15d**, and **16d**) showed preference towards the A isoform, with an hLDHB IC_50_ > 100 μM. The other thirteen hybrids acted as double inhibitors.

For hLDHA, inhibition appears to occur in the active site, based on molecular modelling, while for hLDHB, it takes place in an allosteric site between protein chains.

The compounds reported here can be used for selective inhibition, such as in primary hyperoxalurias, or for double inhibition of both isoenzymes (hLDHA and hLDHB), as seen in certain types of cancer.

In conclusion, this new family of compounds provides a solid foundation for adjusting substituent groups to achieve inhibition for a wide range of applications, depending on the requirement for selective or double inhibition.

## Data Availability

All the data reported are contained within the article and Appendix A.

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
