# Peer review of "Synthesis of Ethyl Pyrimidine-Quinolincarboxylates Selected from Virtual Screening as Enhanced Lactate Dehydrogenase (LDH) Inhibitors"

_ijms, 2024, doi:10.3390/ijms25179744_

Round 1
Reviewer 1 Report
Comments and Suggestions for Authors
The manuscript entitled “Synthesis of ethyl pyrimidine-quinolincarboxylates selected from virtual screening as enhanced lactate dehydrogenase (LDH) inhibitors” (Manuscript Number: ijms-3152455) by Cobo et al. details synthesis of LDH inhibitors based on virtual screening and the evaluation of the compounds for their enzyme inhibitory potential. The following queries need to be addressed before making any decision for the manuscript.
1. Scheme 2, authors should use the arrow which indicates a failed reaction
2. Authors should mention reaction conditions like time, and yields on the schemes
3. Spacing, typographical errors should be corrected in the entire manuscript.
4. Authors should detail their rationale in choosing 4R68
5. Authors should make individual comparison of the ligands in Figures and provide similarities and differences observed with reference to co-crystal.
6. The orientation of the co-crystal and the representation of the active site residues varies in different figures.
7. Can authors provide SAR based on the in silico and in vitro results?
8. The HRMS (ESI-QTOF) (M-H) calc. for C16H11FN2O2 for compound 4 is not matching with the found mass.
Comments on the Quality of English Language
Moderate editing of English language required.
Author Response
Dear Editor and reviewers
We appreciate very much the time dedicated to the revision and comments made by both reviewers that we think all are very appropriate and we have tried to answer all of them. The changes are highlighted in the manuscript in yellow.
Next, you can find thorough answers to all comments; that I will upload to both reviewers so both can see the whole modification done.
Comment 1. Scheme 2, authors should use the arrow which indicates a failed reaction
Answer: The original arrow shown in Scheme 2 (dashed arrow: --->) has been changed for the crossed arrow, as required.
Comment 2. Authors should mention reaction conditions like time, and yields on the schemes
Answer: All schemes have been revised according to this recommendation of referee 1 and, where needed, reaction conditions have been implemented, including not only solvents and temperature, but also reaction times and yields. Table 1 summarizes all the changes made.
Table 1. Changes made in all different schemes.
|
Scheme |
Original information |
Added information |
|
Scheme 1 |
Base, solvent, type of heating, reaction time, yield and product description |
No info has been added. Yield has been modified in the way it is given. Instead of giving a hairpin, it has been specified for each compound |
|
Scheme 2 |
Solvent and heating |
Reaction time. As if did not work, yield is not given. |
|
Scheme 3 |
Reagent, solvent, type of heating, yield and product description |
Reaction time has been included |
|
Scheme 4 |
Solvent, heating, reaction time, yield |
Yields have been specified for each compound, as made for scheme 1. |
|
Scheme 5 |
Solvent, base, reagent, type of heating |
Reaction time and yields have been included in the three reaction steps; and the scheme put in two lines in order to get structures with the same size as the others.
|
Comment 3. Spacing, typographical errors should be corrected in the entire manuscript.
Answer: All detected typographical errors have been corrected, as suggested by the MS-word grammar-corrector. All the hyphens put as - (minus) have been changed to the bigger one: –.
Some paragraphs have been moved around figures or schemes to avoid blanks in the corresponding pages, these are highlighted in yellow.
Comment 4. Authors should detail their rationale in choosing 4R68
Answer: According to this comment, and the one made by referee 2, the original statement “The preliminary virtual screening was performed by molecular docking using hLDHA-W31 complex (pdb code: 4R68), as previously reported [46]” has been modified (see comment 4 of referee 2).
We have analyzed all pdb deposited structures and finally, we decided to go on with 4R68 to perform the docking analysis, because although all the structures have the inhibitors located in the active site, inhibitor W31 in 4R68:
- has a larger structure and occupies more volume to perform the docking, respect to for example 5W8H, which is smaller (commented by ref. 2),
- gives interactions with the four main Aa residues described with the natural ligand and responsible so of its activity, 5W8H for example gives only interactions with two of them.
- has an IC50 in the range of nanomolar, not like 5W8H that is in micromolar, for example.
- its resolution (2.11 Å) is not bad, and other like 5W8H, although with better resolution 1,8 Å, its inhibitor 9Y1 is located in different positions, differently even in every chain with two molecules in one change; so, we do not think this is a reliable structure to take as reference independently to its resolution.
Comment 5. Authors should make individual comparison of the ligands in Figures and provide similarities and differences observed with reference to co-crystal.
Answer: According to ref 1 we are not comparing the placement of the different hybrids with the reference co-crystalized. However, regarding the hLDHA enzyme, throughout the text in the original draft it is stated that:
(…) The better inhibitory results of U-shaped 15 than 13 and 14 is because of their better placement of 15a and 15b in the active site of the enzyme, similar to that of the reference used (W31), whilst (13,14)(a,b) are located outside of the pocket. This is depicted in Figure 6.
(…) Additionally, the vast majority of hybrids (15-18)(a-d) are placed in the active site of the hLDHA enzyme in a similar way to the reference W31. This is exemplified with 18c in Figure 7.
(…) However, there are some hybrids (15b, 16c, 17c and 18a) which are located differently in the enzyme to the way W31 does.
This way, in order to fulfil the requirements of referee 1, we have included (in the supplementary material) the individual comparison of the pose of each ligand with the one of the reference W31.
Regarding the hLDHB enzyme, no comments were made according to their placement.
Thus, we have included a new paragraph indicating the ligands which were located in the allosteric site in a similar way to H1U and those that had similar positioning among them:
(...) we observed that all compounds were located in the allosteric site close to the reference H1U, having compounds 16a, 17a and 18a a very similar placement to H1U. Additionally, compounds 15d, 16b, 16d and 17d had a very similar placement among them.
Additionally, we have included (in the supplementary material) the individual comparison of the pose of each ligand with the one of the reference H1U.
(…) For detailed placement comparison of each compound in the hLDHB allosteric site, see figures S17-S32 in Supplementary Material.
Comment 6. The orientation of the co-crystal and the representation of the active site residues varies in different figures.
Answer: As reviewer 2 requires changes in the figures, this comment should be readdressed consequently.
Comment 7. Can authors provide SAR based on the in silico and in vitro results?
Answer: At this point, it is not clear to make a reliable SAR comparing both isoenzymes, we need to go deep in with more structures and modifications to make a statement for that. There are in the manuscript some SAR comments individually performed for each enzyme but not a correlation between them.
Comment 8. The HRMS (ESI-QTOF) (M-H) calc. for C16H11FN2O2 for compound 4 is not matching with the found mass.
Answer: A writing mistake was made when typing the found mass. It has been checked and properly modified, as shown in the manuscript: HRMS (ESI-QTOF) (M-H) calc. for C16H11FN2O2: 281.0732 found: 281.0732
Comments on the Quality of English Language. Moderate editing of English language required.
Answer: As a consequence of the required language revision, the following changes have been made, and put in red in the manuscript:
- Line 12. To the original sentence “These were (…)” the word inhibitors has been added. New statement: “These inhibitors were (…)”
- Line 13. The word “the” of the original sentence “(…) pathway by the coupling the key ethyl (…)” has been removed. New statement: “(…) pathway by coupling the key ethyl (…)”
- Lines 14-15. The original sentence “(…) prepared by Pfitzinger synthesis and further esterification, (…)” has been changed for the following: “(…) which were prepared by Pfitzinger synthesis followed by a further esterification, (…)”
- Line 16. The word “and” of the original sentence “(…) and in a green solvent (…)” has been removed. New statement: “(…) in a green solvent (…)”
- Line 18. The words “with” and “of them” from the original sentence “(…) thirteen of them with IC50 values (…)” have been replaced by “having” and “hybrids” respectively, and so the sentence keeps as “(…) thirteen hybrids having IC50 values (…)”
- Line 19. The word “hybrids” from the original “(…) all hybrids with IC50 < 10 μM (…)” has been changed by “compounds”.
- Line 29. The word “ones” has been added to the original statement “(…) most important are the (…)” and now is “(…) most important ones are the (…)”
- Lines 30 and 31. Punctuation marks (, and ;) have been added to clarify the original statement “(…) (M4) mostly found in human liver and skeletal muscle and hLDHB (H4) in human (…)”
- Line 33. The original sentence “(…), and one of them that promotes cancer proliferation is known as Warburg effect” has been rewritten to the following: “An example of this is the Warburg effect, which promotes cancer proliferation.”
- Line 34. The original statement “In this, aerobic” has been completed by adding “metabolic alteration” and the word “the” of “ after (…) place over the oxidative (…)” has been removed.
- Line 38. Linker “In addition” has been changed for “In addition to this”
- Line 40. The word “and” of “primary hyperoxalurias (PHs) [18-20] and arthritis and (…)” has been changed for a comma, being the new sentence: “primary hyperoxalurias (PHs) [18-20], arthritis and (…)”
- Line 45. To avoid word repetition, the sentence “also described as” has been replaced by “reported to be”
- Line 47. To avoid word repetition, the sentence “is reported to be involved” has been replaced by “is also involved”.
- Line 54: The original sentence “(…),the molecular inhibition of LDH using small–molecules presents…” has been rewritten: “(…)the LDH inhibition using small–molecules presents…”
- Lines 67 and 68. The original sentence “(…), being some of them reported as hLDHA inhibitors with either pyrimidine or quinolone moiety” has been rewritten: “(…) being some of them (bearing the pyrimidine and/or the quinolone moiety) reported as hLDHA inhibitors.”
- Line 82. Brackets in the original “by 1,4– (and 1,3)–disubstituted” have been removed.
- Lines 82 and 83. Original sentence “(…) moiety in a non–U–shaped disposition ) (13,14)(a,b) and sixteen linked by a 1,2–disubstituted aryl moiety in a U–shaped disposition (15–18)(a–d).” has been fixed by deleting the “)” symbol and by adding commas.
- Line 85. The word “was measured” has been replaced by “has been performed”
- Line 92. The word “to” in the original sentence “(…) access to quinoline–4–carboxylic intermediates” has been replaced by “the”
- Line 93. The word “the” has been added before “quinoline”.
- Line 98. The second comma of the relative clause has been included: “(…) hLDHA inhibitors [46], all structures (…)”
- Line 103. To avoid repetition, the original “(…) the synthesis of the ones having the ester (…)” has been replaced by “(…) the synthesis of those having the ester (…)”
- Line 107. To avoid repetition of the linker, original “Additionally” has been changed for “Moreover”
- Line 117-118. Punctuation marks have been modified, from the original “(…) were once again tested at three sites, (…)” into new“(…) were, once again, tested at three sites: (…)”
- Line 131. “to check” has been replaced by “regarding”
- Lines 132 and 133. The sentence “U–shaped vs non–U–Shaped and then to decide if following with the full series of U–Shaped hybrids (15–18)(a–d), as shown in Figure 4.” has been rewritten as follows: “the U–shaped vs the non–U–Shaped and, afterwards, to decide whether to synthesise or not the full series of U–Shaped hybrids (15–18)(a–d), as shown in Figure 4”
- Line 150. The word “later” has been introduced
- Line 155. The word “agrees” has been replaced by “is in agreement”
- Line 165. The sentence “desired hybrid A (see Scheme 2), but the reaction did not work” has been modified to make it easier to read: “desired hybrid A, but the reaction did not work (see Scheme 2).”
- Lines 169-170. Original “Although in the vast majority of the attempts the reaction did not work as expected, and if worked (EtOH, MW, 150 °C), they got” has been rewritten: “However, in the vast majority of the attempts, the reaction did not work as expected and, if worked (EtOH, MW, 150 °C), they got”
- Line 176. The word “classic” has been changed for “classical”. And the sentence “(…) conditions, that is, the reaction in ethanol with sulphuric acid, but this did (…)” rewritten: “(…) conditions, performing the reaction in ethanol with sulphuric acid; but this did (…)”
- Line 183. The comma from “conditions, and the use of” has been deleted
- Line 185. The word “later” has been included
- Line 188. Preposition “a” has been included
- Line 195. Preposition “of” has been changed for “for”
- Line 196, the last part of the sentence has been modified to reduce the paragraph and so the scheme can enter in the page, and instead or “(…), as shown in Scheme 4.”, now is “(…) (see Scheme 4). “
- Lines 205-206. The sentence “Table 1 displays the all the above structures together with their reaction details” has been modified: “Table 1 displays all the above structures together with their reaction details”
- Line 222. “These results” has been changed for “The results”
- Line 236. “inhibitory behaviour” has been changed for “inhibition data” and numbers “13 and 16” have been changed for “thirteen and sixteen”
- Line 238. The sentence “(…) and interactions with Arg168” has been modified as follows: “(…) as well as moderate interactions with Arg168”
- Line 261. “the establishment of” has been added
- Line 264. The sentence “what enables a stronger interaction with” has been replaced by “what makes them interact stronger with”
- Line 265. The sentence “(…) or the establishment of other type (…)” has been modified as follows: “(…) or to establish other type (…)”
- Line 271. The pronoun “the” has been replaced by “their” before inhibitory….
- Line 286. Linker “So” has been changed by “In that regard”
- Line 303. “Due to” has been replaced by “a consequence of” and the comma after “hybrids” has been replaced by “;”
- Line 305. “Consequently” has been changed for “Thus” to avoid repetition.
- Line 321. The sentence “(…) there is a reported a selective inhibitor of hLDHB” has been fixed: “(…) there is a reported selective inhibitor of the hLDHB (…)”
- Lines 338-339. The sentence “reference H1U, in similar order (…)” has been modified: “(…) reference H1U, it is of similar order to the ones shown with the different aminoacid residues in the case of hLDHA”
- Line 342. “Either” has been changed for “neither”. Additionally, “of the two possible” has been added.
- Lines 962-964. Paragraph has been rewritten to make it easier to understand. Original: “As a result, we got thirteen of these compounds with IC50 < 5 μM and four of them (16a, 18b, 18c and 18d) with IC50 ≈ 1 μM, which is an improvement of 20 times average in their inhibition respect of the previous family.” New: “As a result, we got thirteen of these compounds with IC50 < 5 μM and four of them (16a, 18b, 18c and 18d) with IC50 ≈ 1 μM, which is a 20 times average improvement in their inhibitory activity when compared to the previous family.”
- Line 972. The word “kinds” has been replaced for “types” before “of cancer…”
Reviewer 2 Report
Comments and Suggestions for Authors
Title and Abstract = The title appropriately reflects the content of the work. The abstract is brief and contains enough information for the reader to understand the principal findings of the study. However, I want to suggest the authors incorporate some sentences about what is the pharmacological usefulness of inhibiting hLDHA.
Introduction = It is easy to read. The authors review the literature and state there are different hLDHA reported to date. However, there is no figure which summarizes some of the molecules they mention in references 8,24,26,41–44. I recommend the incorporation of a Figure with that information and the IC50 (if any) of the compounds.
Figures, general recommendation: If the molecules are drawn with ChemDraw software, I recommend applying the "ACS Document 1996" Style to all the molecules. It will improve the quality and readability of the compounds.
Docking = According to Uniprot there are dozens of PDB available of hLDHA. Why did the authors choose the PDB: 4R68? Nowadays other PDB with better resolution are available (e.g.: 5W8H). A clarificatory statement should be added, the authors used that PDB in previous work (ref. 46), but it is not enough justification.
Docking images 5-9 = In my opinion these images are the weak point of the article. Today is unacceptable to yield low-quality docking interaction images, there are several softwares programs that could be used to give a high-quality result. The interactions, conformation, and superpositions are not clear. Also, there are no 2D diagrams, there is no Table with the docking Scoring (including the W31 compound for comparison). The docking images section must be redone in order to give clear information, other way this section takes away merit from the work.
Author Response
Dear Editor and reviewers
We appreciate very much the time dedicated to the revision and comments made by both reviewers that we think all are very appropriate and we have tried to answer all of them. The changes are highlighted in the manuscript in yellow.
Next, you can find thorough answers to all comments; that I will upload to both reviewers so both can see the whole modification done.
Answers to reviewer 2.
Comment 1. Title and Abstract = The title appropriately reflects the content of the work. The abstract is brief and contains enough information for the reader to understand the principal findings of the study. However, I want to suggest the authors incorporate some sentences about what is the pharmacological usefulness of inhibiting hLDHA.
Answer: According to what has been suggested by ref 2, apart from the changes related to the language edition, an additional sentence has been added to the original abstract, without exceeding the 200 words limitation:
“The molecular inhibition of the hLDHA enzyme has been demonstrated to be of extreme importance in the treatment of cancer and other diseases (such as primary hyperoxalurias).”
Comment 2. Introduction = It is easy to read. The authors review the literature and state there are different hLDHA reported to date. However, there is no figure which summarizes some of the molecules they mention in references 8,24,26,41–44. I recommend the incorporation of a Figure with that information and the IC50 (if any) of the compounds.
Answer: Figure 1 has been added with seven examples of double, hLDHA selective and hLDHB selective inhibitors, as suggested. Consequently, some paragraphs have been moved to avoid blank spaces on some pages due to the size of the schemes.
Comment 3. Figures, general recommendation: If the molecules are drawn with ChemDraw software, I recommend applying the "ACS Document 1996" Style to all the molecules. It will improve the quality and readability of the compounds.
Answer: All figures and schemes have been updated with the new style recommended.
Comment 4. Docking = According to Uniprot there are dozens of PDB available of hLDHA. Why did the authors choose the PDB: 4R68? Nowadays other PDB with better resolution are available (e.g.: 5W8H). A clarificatory statement should be added, the authors used that PDB in previous work (ref. 46), but it is not enough justification.
Answer: We have analyzed all pdb deposited structures and finally, we decided to go on with 4R68 to perform the docking analysis, because although all the structures have the inhibitors located in the active site, inhibitor W31 in 4R68:
- has a larger structure and occupies more volume to perform the docking, respect to for example 5W8H, which is smaller (commented by ref. 2),
- gives interactions with the four main Aa residues described with the natural ligand and responsible so of its activity, 5W8H for example gives only interactions with two of them.
- has an IC50 in the range of nanomolar, not like 5W8H that is in micromolar, for example.
- its resolution (2.11 Å) is not bad, and other like 5W8H, although with better resolution 1,8 Å, its inhibitor 9Y1 is located in different positions, differently even in every chain with two molecules in one change; so, we do not think this is a reliable structure to take as reference independently to its resolution.
Comment 5. Docking images 5-9 = In my opinion these images are the weak point of the article. Today is unacceptable to yield low-quality docking interaction images, there are several softwares programs that could be used to give a high-quality result. The interactions, conformation, and superpositions are not clear. Also, there are no 2D diagrams, there is no Table with the docking Scoring (including the W31 compound for comparison). The docking images section must be redone in order to give clear information, other way this section takes away merit from the work.
Answer: A 2D interaction diagram has been included for compound 18c as figure 7b. Figure 8 has been replaced by the corresponding 2D interaction diagram as, for us, it is the easiest way to explain the different placement of compound 15b when compared to 16c, 17c and 18c as well as to show the different interactions established. 15b interacts only with Arg168 but in a much stronger way than the others, which establish other additional interactions.
Docking score for reference W31 (in case of LDHA) and H1U (in case of LDHB) are already noted in the main text (lines 109 and 328, respectively). Additionally, detailed parameters for EACH compound, including docking score and interactions, are given in Supplementary Material (S4 for LDHA and S6 for LDHB).
We have tried to increase the quality of figures 6 and 7 (prepared for 30 x 20 cm and 300 dpi resolution) and inserted the 2D interactions diagrams, with large fonts and exported for a size of 20 X 24 cm at 300 dpi resolution, if this not enough a clear address of what is not clear of what minimum of quality is required would be appreciated.
Reviewer 3 Report
Comments and Suggestions for Authors
In this article by Cobo and co-workers, the authors describe the synthesis and evaluation of a novel family of ethyl pyrimidine-quinolinecarboxylate derivatives as lactate dehydrogenase (LDH) inhibitors (e.g., against hLDHA and hLDHB). The molecules were selected from virtual docking screening, and then prepared through Pfitzinger synthesis and further reactions. After careful reading of the manuscript, I recommend major revisions.
First of all, I would like to apologize, I reviewed the first versions. I noticed afterwards that there was the corrected one on the mdpi portal.
After checking the new version, I can see that many of my concerns were corrected. However, I have few other comments to help the authors improve the quality of their work.
The publication must be proofread by a native English person as elsewhere in the manuscript the text is sometimes badly expressed. Furthermore, some paragraphs are sometimes difficult to understand, or incomprehensible.
‒ Figure 2: NH2 of compounds 7‒12 must be subscripted.
‒ Figure 2: the shadow of the rectangles should be removed. Maybe replaced by dotted rectangles.
‒ Figure 2: This work, second reaction. Can we add a name for this reaction (Buchwald Hartwig, Buchwald Hartwig like reaction??).
‒ L132: italicize vs and put a dot (see also elsewhere in the article).
‒ L145: aminoacetophenones
‒ L161, 163, 181, 192: It is indeed an aminolysis reaction, but the term is confusing.
‒ L164: stablished or established?
‒ Every figures, schemes including molecular structures should be drawn using a similar template, size…(i.e., Fig. 1,2,3,4 vs. Scheme 1,2,3,4,5 and table 1).
‒ Every multiply signs must be inserted via the insert symbol command (see elsewhere in the manuscript).
‒ Materials and Methods (L361): Leave a blank line after otherwise, and before Melting.
‒ L394: space after 254.
‒ Every melting points should be in °C.
‒ L429: subscript f of Rf.
‒ 1 of 1H and 13 of 13C must be superscript (L430, L432).
‒ L705: space after 170.
‒ L721: Mr ?
Otherwise, the overall investigation was performed adequately.
Comments on the Quality of English LanguageThe publication must be proofread by a native English person
Author Response
Thanks very much the reviewer for taking the time to review this manuscript to improve it. Here next you can find a detailed response to your queries.
Query: The publication must be proofread by a native English person as elsewhere in the manuscript the text is sometimes badly expressed. Furthermore, some paragraphs are sometimes difficult to understand, or incomprehensible.
Answer: A second major English edition, as suggested by this reviewer, has been carried out by a native English speaker, who has been included in the acknowledgments. The changed paragraphs have been highlighted in green in the manuscript. In addition to changing some connectors to clear the sentence, the edition has consisted of splitting long sentences into two or even three connected sentences to make them understandable; the manuscript is much clearer understood after this second thorough English edition
Query: Figure 2: NH2 of compounds 7‒12 must be subscripted.
Answer: Because of the format change, it was not subscripted. Now it has been done.
Query: Figure 2: the shadow of the rectangles should be removed. Maybe replaced by dotted rectangles.
Answer: The rectangles have been changed.
Query: Figure 2: This work, second reaction. Can we add a name for this reaction (Buchwald Hartwig, Buchwald Hartwig like reaction??).
Answer: The Buchwald-Hartwig Cross Coupling Reaction is defined as a C-N coupling reaction involving halogenated aromatic hydrocarbons and aromatic amines in the presence of a palladium catalyst, providing a new synthetic route to create amine-linked CMPs with strong redox properties. So, this is not our case, in which there is no catalyst and it is a simply aromatic nucleophilic substitution; we do not think this name fits here.
Query: L132: italicize vs and put a dot (see also elsewhere in the article).
Answer: It has been done. Lines 132 and 932, it has been highlighted in yellow over the green
Query: L147: aminoacetophenones
Answer: The missing “n” has been added, highlighted in blue.
Query: L161, 163, 181, 192: It is indeed an aminolysis reaction, but the term is confusing.
Answer: This is a well-known term that describes the replacement of a halogen (or other leaving group) in an alkyl/aryl group (R−X) by an amine (R'−NH2) with the elimination of hydrogen halide (HX), although it is more used in the reaction of carboxylic derivatives with amines to give the corresponding amide, and that might confuse the reviewer.
This is even defined in the general wikipedia (https://en.wikipedia.org/wiki/Aminolysis) to exemplify the well-established it is, this term entitles some papers It is not worth making a list of them and the dictionaries in which this term is defined.
We would like to keep this term but, If the reviewer and editorial board think this is not appropriate it could be changed into nucleophilic substitution; we have modified lines L-144 and L-145, in which first appears it to help introducing the term and so written now as
“Subsequently, the aminophenylquinoline residue would be coupled with the chloropyrimidine fragment through an aromatic nucleophilic substitution (aminolysis from here)”
This is highlighted in yellow
Query: L164: stablished or established?
Answer: the right term is of course “established”, sorry about the archaism. It has also been fixed in the text elsewhere detected or new introduced, or now introduced in the edition.
Query: Every figures, schemes including molecular structures should be drawn using a similar template, size…(i.e., Fig. 1,2,3,4 vs. Scheme 1,2,3,4,5 and table 1).
Answer: As the size of any scheme or figure is different when copy from Chemdraw to MS-word, even having the same template (ACS), the size is automatically adapted. In order to keep the same size for all the structures, we have readapted all figures and schemes and put the scale of the bigger one in the manuscript (figure 2, with 91%) that ensure every figure/scheme have same size for structures. So, this query has been accomplished.
Query: Every multiply signs must be inserted via the insert symbol command (see elsewhere in the manuscript).
Answer: All multiply signs detected in the manuscript has been changed accordingly and highlighted in blue. L376, L409, L551, L725;
Query: Materials and Methods (L361): Leave a blank line after otherwise, and before Melting.
Answer: a blank line has been added
Query: L394: space after 254.
Answer: Done
Query: Every melting points should be in °C.
Answer: Done
Query: L429: subscript f of Rf.
Answer: Done
Query: 1 of 1H and 13 of 13C must be superscript (L430, L432).
Answer: Done, additionally the d6 of DMSO has been italicized
Query: L705: space after 170.
Answer: Done
Query: L721: Mr ?
Answer: This is an accepted abbreviation for molecular mass of a substance, formerly also called molecular weight that can be also abbreviated as MW,
In addition, we have edited some pages moving paragraphs, to avoid blanks because of the size of the corresponding figures or schemes
Round 2
Reviewer 1 Report
Comments and Suggestions for Authors
Comments have been addressed
Comments on the Quality of English LanguageNone
Author Response
Thanks again the reviewer for taking the time to review this manuscript to improve it. A second major English edition made by a native English speaker has been carried out, as suggested by reviewer 3. The changed paragraphs have been highlighted in green in the manuscript. It is not anything to respond to, as no more requests were made
Reviewer 2 Report
Comments and Suggestions for Authors
Authors have addressed the issues arised and now the papers has improved a lot. I recommend it for final publication.
Author Response

(The authors gave the same response as above.)

Reviewer 3 Report
Comments and Suggestions for Authors
In this article by Cobo and co-workers, the authors describe the synthesis and evaluation of a novel family of ethyl pyrimidine-quinolinecarboxylate derivatives as lactate dehydrogenase (LDH) inhibitors (e.g., against hLDHA and hLDHB). The molecules were selected from virtual docking, and then prepared through Pfitzinger synthesis and further reactions. After careful reading of the manuscript, I recommend minor revisions, and publication in IJMS.
The introduction include sufficient background and relevant references, the research was designed appropriately, and the results were clearly presented. Furthermore, the publication was proofread by a native English person, although the way the manuscript is written may sometimes be a little clumsy.
See below for the minor revisions:
L62 : compounds
L75 : pyrimidine-quinolones or pyrimidine-quinolone ?
Every dash in Fig. 2 (e.g., 7‒12, a‒d, etc.), Fig 4, scheme 1,3,4,5, Table 1 must be inserted via the insert symbol command like it was performed in the text (take from word document and copy paste in chemdraw).
L1219-1220: see template for the journal abbreviation (i.e., dots)
L1243: see template for the journal abbreviation
L1248: see template for the journal abbreviation
Author Response
Thanks very much the reviewer for taking the time and thoroughness to review this manuscript again to detect some mistake that needs minor revisions. Here, next you can find a detailed response to your queries, which in the text have been yellow highlighted
Request: L62 : compounds
Answer: Misspelled word compounds has been properly written.
Request: L75 : pyrimidine-quinolones or pyrimidine-quinolone ?
Answer: The missing s has been included. Pyrimidine-quinolones was the appropriate one
Request: Every dash in Fig. 2 (e.g., 7‒12, a‒d, etc.), Fig 4, scheme 1,3,4,5, Table 1 must be inserted via the insert symbol command like it was performed in the text (take from word document and copy paste in chemdraw).
Answer: All hyphen symbols (-) have been replaced by the appropriate symbol (─), as suggested.
Request: L1219-1220: see template for the journal abbreviation (i.e., dots)
Answer: The journal abbreviation has been corrected and dots included, now is Pharm. Chem. J.
Request:L1243: see template for the journal abbreviation
Answer: The journal abbreviation has been corrected and dots included, now is Acta Crystallogr. C
L1248: see template for the journal abbreviation
Answer: The journal abbreviation has been corrected and dots included, now is J. Appl. Crystallogr.